# Exactly Computing do-Shapley Values

**R. Teal Witter** [* 1]   **Álvaro Parafita** [* 2]   **Tomàs Garriga** [2 3]   **Maximilian Muschalik** [4 5]   **Fabian Fumagalli** [4 5]
**Axel Brando** [2]   **Lucas Rosenblatt** [6]

## Abstract

Structural Causal Models (SCM) are a powerful framework for describing complicated dynamics across the natural sciences. A particularly elegant way of interpreting SCMs is via do-Shapley values, a game-theoretic method of quantifying the average effect of $d$ variables across exponentially many interventions. Like Shapley values, computing do-Shapley values generally requires evaluating exponentially many terms. The foundation of our work is a reformulation of do-Shapley values in terms of the *irreducible sets* of the underlying SCM. Leveraging this insight, we can exactly compute do-Shapley values in time linear in the number of irreducible sets $r$, which itself can range from $d$ to $2^d$ depending on the graph structure of the SCM. Since $r$ is unknown a priori, we complement the exact algorithm with an estimator that, like general Shapley value estimators, can be run with any query budget. As the query budget approaches $r$, our estimators can produce more accurate estimates than prior methods by several orders of magnitude, and, when the budget reaches $r$, return the Shapley values up to machine precision. Beyond computational speed, we also reduce the identification burden: we prove that non-parametric identifiability of do-Shapley values requires only the identification of interventional effects for the $d$ singleton coalitions, rather than all classes. Our code is available here.[1]

## 1. Introduction

The question of causality is crucial to scientific inquiry, ranging from policy evaluation in economics to treatment effects

---
[1]Claremont McKenna College [2]Barcelona Supercomputing Center [3]Novartis [4]LMU Munich [5]MCML [6]Williams College. Correspondence to: R. Teal Witter <rtealwitter@cmc.edu>, Álvaro Parafita <parafita.alvaro@gmail.com>.

*Proceedings of the 43rd International Conference on Machine Learning*, Seoul, South Korea. PMLR 306, 2026. Copyright 2026 by the author(s).

[1]github.com/rtealwitter/exactdoshap

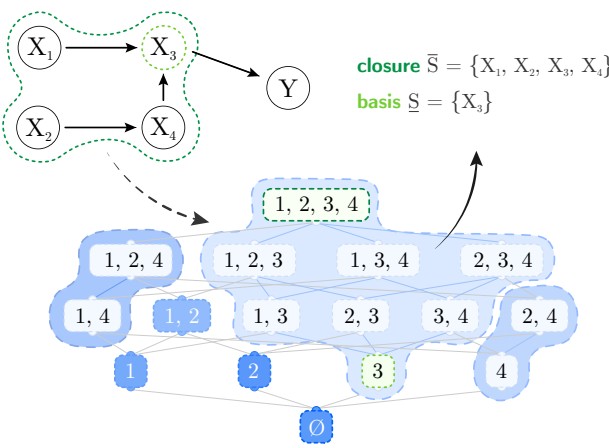

**Structural Causal Model (SCM)**

closure $\bar{S} = \{X_1, X_2, X_3, X_4\}$
basis $\underline{S} = \{X_3\}$

**Lattice of Coalitions**

*Figure 1.* An example Structural Causal Model (SCM) and the corresponding lattice of coalitions. Because of the graph structure, intervening on some nodes is redundant. For example, setting $\{X_1, X_2, X_3, X_4\}$ has the same effect as setting $\{X_3\}$ because $X_3$ blocks all directed paths from the other nodes to $Y$. For such a *class*, we refer to its smallest coalition (e.g., $\{X_3\}$) as the *basis*, and the largest coalition (e.g., $\{X_1, X_2, X_3, X_4\}$) as the *closure*.

in healthcare. Yet, observational data alone is often insufficient due to the fundamental problem of causal inference: because we cannot observe the counterfactual world where a specific intervention did *not* occur, we cannot definitively state, based on data alone, that one event *caused* another (Holland, 1986; Rubin, 1974).

Structural Causal Models (SCMs) offer a powerful solution by explicitly modeling the underlying mechanisms of a system (Pearl, 2009). Whether derived from established domain knowledge or learned via causal discovery algorithms (Peters et al., 2017), SCMs formalize the data-generating process: a directed acyclic graph $G$ representing causal relationships, and a set of structural equations that determine the value of each node as a function of its parents and exogenous noise (Bareinboim & Pearl, 2016). We provide a more formal introduction to SCMs in Appendix C.

With a fully specified SCM, we can rigorously evaluate the effect of specific actions using the *do*-operator (Pearl, 2009). This operator simulates an intervention where a subset of

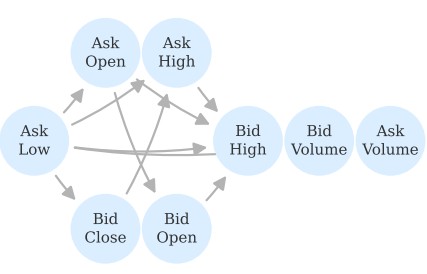
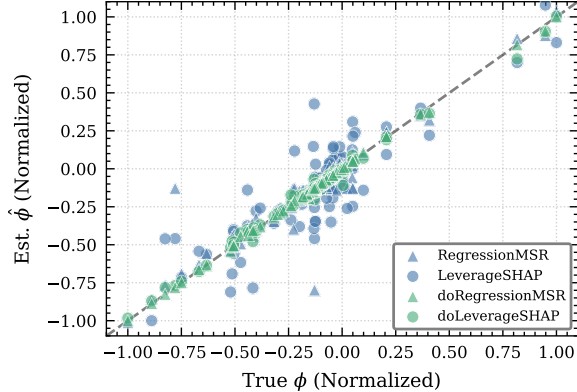

*Figure 2.* **Left: A learned SCM** from a TALENT dataset. Nodes and edges represent the learned causal graph used to define the interventional value function $\nu(S) = \mathbb{E}[Y \mid \mathrm{do}(S = \mathbf{x}_S)]$ for a fixed instance $\mathbf{x}$. **Right: Plot of estimated vs true do-Shapley values** on the learned SCM for randomly sampled $\mathbf{x}$. Compared to the value-function-agnostic state-of-the-art `RegressionMSR` and `LeverageSHAP` estimators, our *doEstimator* variants provide substantially more accurate do-Shapley value estimates.

variables is forced to take specific values, independent of their natural causes. Consider a specific instance of interest $\mathbf{x} \in \mathbb{R}^d$. We define the value function $\nu(S)$ as the expected value of the target outcome $Y$ when the subset of features $S \subseteq [d]$ is intervened upon to match their observed values in $\mathbf{x}$:

$$\nu(S) = \mathbb{E}[Y \mid \mathrm{do}(S = \mathbf{x}_S)]. \tag{1}$$

This formulation enables us to precisely answer hypothetical queries, such as: *"If we explicitly set this student's income and tutoring time, how would their probability of admission change?"* or *"If a patient were administered prednisone and made to stop smoking, what would be their expected pain level?"* However, characterizing the system's behavior purely through these individual queries is challenging. As the number of features $d$ grows, the number of possible interventional subsets scales as $2^d$. To extract interpretable insights from this combinatorial landscape, we need a unified framework to attribute the complicated dynamics of the SCM to individual features.

The Shapley value (Shapley, 1953) provides a rigorous framework for such explanations by attributing the changes in the outcome $Y$ to individual variables based on their marginal contributions. Formally, the $i$th Shapley value captures the weighted average effect of adding variable $i$ to a coalition $S$:

$$\phi_i = \sum_{S \subseteq [d] \setminus \{i\}} [\nu(S \cup \{i\}) - \nu(S)] p_{|S|} \tag{2}$$

where the weight $p_\ell = \frac{1}{d}\binom{d-1}{\ell}^{-1}$ can be interpreted as a probability distribution.

When the value function is defined via the interventional *do*-operator (Equation 1), the result is the *do-Shapley value*

(Jung et al., 2022), also referred to as the causal Shapley value (Heskes et al., 2020).[2] Unlike standard formulations that rely on conditional expectations (Lundberg & Lee, 2017) or restrictive path-dependent permutations (Frye et al., 2020), this metric strictly isolates the total causal effect of a feature intervention. This rigorous isolation allows us to translate abstract model dynamics into concrete causal attributions, such as statements like *"High family income increased acceptance probability by $10\%$"* or *"Prescribing prednisone decreased reported pain by two marks."*

The challenge in computing the Shapley value, of course, is that there are still $2^d$ terms $\nu(S)$. So, without additional structure in $\nu$, exactly computing the Shapley value would take exponential time. To address this, the standard approach is to approximate the Shapley value using stochastic estimators that evaluate $\nu(S)$ on a limited budget of sampled coalitions. A diverse array of model-agnostic estimators has been developed for this purpose, including direct Monte Carlo estimators (Štrumbelj & Kononenko, 2014), permutation-based sampling (Castro et al., 2009), and regression-based formulations such as `KernelSHAP` and `LeverageSHAP` (Lundberg & Lee, 2017; Covert & Lee, 2021; Musco & Witter, 2025). We discuss additional related works in Appendix B.

For do-Shapley values, recent work has exploited the observation that the topological structure of SCMs often renders specific interventions redundant (Parafita et al., 2025). For example, in the causal graph depicted in Figure 1, the intervention on $\{X_3\}$ results in the same value as the intervention on $\{X_1, X_3, X_4\}$, because the paths from $X_1$ and $X_4$ to $Y$ are blocked by $X_3$. To formalize this, define the *basis* of a

---

[2]For conciseness, we henceforth refer to the do-value function simply as the value function, and the do-Shapley value as the Shapley value.

coalition $S$ as the subset $\underline{S} \subseteq S$ containing precisely those variables $j \in S$ that possess a directed path to $Y$ that does *not* traverse any other node in $S$. In effect, any variable in $S \setminus \underline{S}$ is intercepted by $\underline{S}$ and yields no additional impact on the outcome, ensuring $\nu(S) = \nu(\underline{S})$. A set is *irreducible* if it is its own basis. This property enables a caching strategy: rather than naively evaluating the SCM for every query $\nu(S)$, the estimator first computes the basis $\underline{S}$ and checks if $\nu(\underline{S})$ has been previously memoized. If so, the cached value is returned; if not, the SCM is evaluated and the result stored. This mechanism avoids redundant evaluations of the underlying model, resulting in significant computational speedups (Parafita et al., 2025).

We extend this insight by observing that every basis $\underline{S}$ is associated with a unique *closure* $\overline{S} \supseteq \underline{S}$—the maximal coalition such that intervening on $\overline{S}$ yields the identical effect as intervening on $\underline{S}$. For example, the closure of $\{X_3\}$ in Figure 1 is $\{X_1, X_2, X_3, X_4\}$. Together, these bounds define an equivalence class of coalitions $\{S \subseteq [d] : \underline{S} \subseteq S \subseteq \overline{S}\}$, all of which map to the same value $\nu(\underline{S})$.

Crucially, these classes constitute a partition of the powerset of all $d$ features. Letting $r$ denote the total number of such classes $c_1, \ldots, c_r$, we leverage this structure to compress the Shapley summation into a linear combination of class values:

$$\phi_i = \sum_{j=1}^{r} \nu(c_j) w_i(c_j) \qquad (3)$$

where the class weight $w_i(c_j)$ is derived in Equation 5. A similar decomposition is known for trees (Zern et al., 2023b; Witter et al., 2025); but, unlike trees where the structure can be read in a linear pass of the leaves, efficiently finding the class structure of an SCM is non-trivial.

This formulation reduces the exact computation of Shapley values to a sum over $r$ terms. To identify these classes efficiently, we propose a lattice exploration algorithm that circumvents the exhaustive enumeration of the powerset. Leveraging a structural property of closed sets (Lemma 3.1), our algorithm enumerates all $r$ classes in $O(r(d + e + T))$ time, where $e$ is the number of edges in the causal graph and $T$ is the time to query the value function once.

The efficiency of this approach is strictly governed by the underlying graph topology. As illustrated in Figure 3, the number of classes $r$ varies with the graph structure, ranging from a linear $d$ in the best case to $2^d$ in the worst case.

To quantify how much this compression manifests in practice, we plot the number of irreducible sets $r$ against the number of variables $d$ across real datasets in Figure 4. The observed scaling typically lies below the worst-case $2^d$, reflecting the sparsity of learned causal graphs in real-world tabular domains.

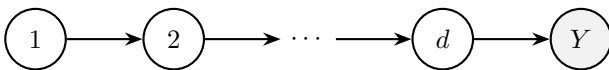

*(a)* Since there is a directed path from each node to $Y$, there are at least $d$ irreducible sets. The figure depicts a graph with $d$ irreducible sets.

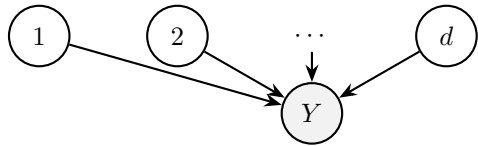

*(b)* Since there are $2^d$ sets total, there are at most $2^d$ irreducible sets. The figure depicts a graph with $2^d$ irreducible sets.

*Figure 3.* The number of irreducible sets ranges between $d$ and $2^d$.

Although our exact algorithm scales linearly with $r$, the number of classes $r$ is unknown a priori. Since real-world applications demand strict resource limits, we must often operate within a fixed computational budget of $m$ value function queries. To address this, we propose a class of estimators explicitly tailored to this constrained setting.

The fundamental limitation of prior caching-based approaches is *sample redundancy*. Standard estimators sample coalitions without knowledge of the underlying causal structure, meaning they can (wastefully) query different coalitions that belong to the same large equivalence class. Thus, a budget of $m$ queries often produces far fewer than $m$ unique values. We resolve this inefficiency by introducing a *boundary sampler*, a targeted exploration strategy guaranteed to identify $\min(m, r)$ distinct equivalence classes when run with $m$ queries. By feeding these distinct values into a simulated estimator, we maximize the information extracted from the available budget. We find that this method can reduce estimation error by orders of magnitude compared to the best value-function-agnostic estimators run with the caching scheme of Parafita et al. (2025). Furthermore, the

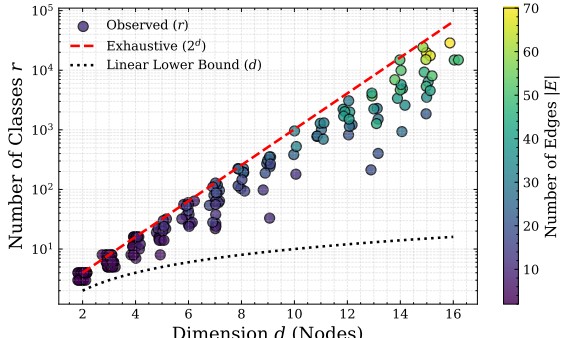

*Figure 4.* **Complexity Reduction.** The number of irreducible sets $r$ (colored points) versus the dimension $d$. While the theoretical worst-case complexity is $2^d$ (red dashed line), real-world causal structures are often sparse, resulting in $r$ scaling in between the exponential and the linear lower bound $d$ (black dotted line).

estimator exhibits seamless convergence: as the budget $m$ approaches $r$, the approximation error vanishes, achieving exact computation (up to machine precision) once $m \geq r$.

In practice, SCMs are frequently learned from observational data and a hypothesized graph structure. A critical prerequisite for estimation through SCMs is *identifiability*: whether a causal query can be uniquely estimated from the observed probability distribution given the graph (Pearl, 2009). For instance, in the presence of unobserved confounding, different structural parameters could yield the exact same observational distribution but different interventional outcomes. The gold standard for verifying non-parametric identifiability is the ID algorithm (Shpitser & Pearl, 2006; Tian & Pearl, 2002), which determines if a specific query $\nu(S)$ is computable from the observational distribution.

This creates a practical bottleneck for Shapley value estimation. Since the Shapley value aggregates $r$ terms, a practitioner using prior methods would be forced to run the estimator and iteratively check identifiability for each irreducible set encountered. If a single coalition proved unidentifiable, the entire estimation would be invalidated after significant computation (Parafita et al., 2025). In Section 5, we resolve this burden by proving a structural sufficiency theorem: to guarantee the identifiability of *all* $2^d$ coalitions, it suffices to verify identifiability for only the $d$ singleton interventions $\{i\} \subset [d]$. This allows practitioners to run a rapid, $O(d)$ sanity check before model training begins, ensuring that the resulting Shapley values will be valid without the risk of costly mid-computation failures.

In summary, our contributions are three-fold:

**1. Exact Computation via Irreducible Sets:** We propose an algorithm that computes exact do-Shapley values by exploiting the graph's equivalence classes. By traversing the lattice of closed sets, the algorithm runs in time linear in the number of irreducible sets $r$ (and the graph size $e$), rather than the worst-case exponential complexity of $2^d$.

**2. Structure-Aware Estimation:** We introduce a class of boundary sampling estimators designed for fixed-budget settings. Unlike prior methods that sample blindly, our approach targets distinct equivalence classes. This yields error reductions of several orders of magnitude as $m$ approaches $r$, and transitions to exact machine-precision computation once the budget satisfies $m \geq r$.

**3. Efficient Identifiability Check:** We prove that non-parametric identifiability of the full do-Shapley value is guaranteed if and only if the $d$ singleton interventions are non-parametrically identifiable. This result enables a rapid sanity check, allowing practitioners to verify the feasibility of the explanation task before incurring the cost of model training or estimation.

While we focus on the Shapley value due to its widespread adoption, our theoretical insights generalize to the broader class of probabilistic values—e.g., Banzhaf values (Banzhaf III, 1964), beta Shapley values (Kwon & Zou, 2022), and weighted Banzhaf values (Li & Yu, 2024)—by simply changing the marginal weights in Equation 4. Furthermore, in Appendix I, we demonstrate how to adapt our methods to compute Shapley Interaction Indices (Grabisch, 1997), capturing the joint causal impact of coalitions.

## 2. Reformulating do-Shapley Values

In this section, we leverage the underlying structure of an SCM to reformulate do-Shapley values in terms of equivalence classes.

Firstly, we will assume that all nodes in $[d]$ are ancestors of the target node $Y$, since non-ancestors have null do-Shapley value. We will start with the definition of basis, related to the concept of irreducible sets in Parafita et al. (2025).

**Definition 2.1** (Basis). The *basis* of $S$, denoted $\underline{S}$, is the subset of nodes $j \in S$ such that there exists a directed path from $j$ to $Y$ that intersects $S$ only at $j$.

A set $S \subseteq [d]$ is *irreducible* if it is its own basis.

Similarly, we will define the closure of $S$ as the superset of nodes that can be blocked from reaching $Y$ by $S$.

**Definition 2.2** (Closure). The *closure* of $S$, denoted $\overline{S}$, is the set of all nodes $j \in [d]$ such that every directed path from $j$ to $Y$ intersects $S$.

We say a set $S \subseteq [d]$ is *closed* if it is its own closure.

Let $\underline{S}$ and $\overline{S}$ be the basis and closure of a coalition $S$, respectively. $S$ belongs to an *equivalence class* with all $T$ such that $\underline{S} \subseteq T \subseteq \overline{S}$. By definition, all nodes in $T \setminus \underline{S}$ are *blocked* from reaching $Y$ by $\underline{S}$, so, by the third rule of do-Calculus (Pearl, 2009),

$$\nu(\underline{S}) = \nu(T) = \nu(\overline{S}).$$

See Appendix D for more details. Note also that all classes form a partition of the $2^d$ coalitions.

Let $r$ be the number of irreducible sets, and denote the classes by $c_1, \ldots, c_r$. In an abuse of notation, we will define $\nu(c_j) = \nu(S)$ where $S$ is any set in class $c_j$. We will use this structure to rewrite the Shapley values:

---

**Algorithm 1** `FindClass`

---

**Input:** Set $S \subseteq [d]$, graph $G$
**Output:** basis $\underline{S} \subseteq S$, and closure $\overline{S} \supseteq S$
$G' \leftarrow G$ with all incoming edges to nodes in $S$ removed
$N_{\text{anc}} \leftarrow$ ancestor nodes i.e., a directed path to $Y$ in $G'$
$\overline{S} \leftarrow S \cup ([d] \setminus N_{\text{anc}})$ ▷ $S$ and nodes not connected to $Y$
$\underline{S} \leftarrow S \cap N_{\text{anc}}$      ▷ Subset of $S$ still connected to $Y$
**return** $\underline{S}, \overline{S}$

---

$$\phi_i = \sum_{S \subseteq [d] \setminus \{i\}} p_{|S|} \left[ \nu(S \cup \{i\}) - \nu(S) \right]$$

$$= \sum_{S \subseteq [d]} \nu(S) \left[ \mathbb{1}[i \in S] p_{|S|-1} - \mathbb{1}[i \notin S] p_{|S|} \right]$$

$$= \sum_{j=1}^{r} \nu(c_j) \cdot w_i(c_j) \tag{4}$$

where, with $\overline{S}$ as the closure of class $c$ and $\underline{S}$ as the basis of class $c$, we define

$$w_i(c) = \sum_{T : \underline{S} \subseteq T \subseteq \overline{S}} \left[ \mathbb{1}[i \in T] p_{|T|-1} - \mathbb{1}[i \notin T] p_{|T|} \right].$$

Even though there could be exponentially many subsets in a class, we can compute $w_i(c)$ in $O(d)$ time. In particular, it is easy to show that

$$
w_i(c) = \begin{cases}
\sum_{\ell=|\underline{S}|}^{|\overline{S}|} p_{\ell-1} \binom{|\overline{S}|-|\underline{S}|}{\ell-|\underline{S}|} & i \in \underline{S} \\[2mm]
-\sum_{\ell=|\underline{S}|}^{|\overline{S}|} p_{\ell} \binom{|\overline{S}|-|\underline{S}|}{\ell-|\underline{S}|} & i \notin \overline{S} \\[2mm]
0 & \text{else.}
\end{cases} \tag{5}
$$

We will use this structure to compute Shapley values. If we have $O(r(d+e))$ time, then we can exactly compute Shapley values as described in Section 3. Since $r$ is initially unknown, we may also want to estimate Shapley values given a fixed query budget $m$. In Section 4, we describe estimators that run in $O(m(d+e))$ time.

## 3. Exactly Computing do-Shapley Values

By Equation 4, we can exactly compute the Shapley value if we know all the irreducible sets. It remains to find all irreducible sets.

A naive strategy is to traverse the set lattice by subset size, determining the class of each set via Algorithm 1. Of course, there are $2^d$ sets on the lattice, so even constant work per set is infeasible. Instead, we can *efficiently* traverse the set

lattice by only generating sets which are guaranteed to be closed, amortizing the work to each class, rather than each subset. The key tool is a structural lemma on an alternate definition of closed sets.

**Lemma 3.1.** *Let $\overline{S} \subset [d]$ be a closed set with basis $\underline{S}$. Then*

1. *For all $j \in \underline{S}$, $\overline{S} \setminus \{j\}$ is closed.*

2. *If $\overline{S} \neq [d]$, there exists $j \in [d] \setminus \overline{S}$ so that $\overline{S} \cup \{j\}$ is closed and $j$ is in the basis of $\overline{S} \cup \{j\}$.*

We defer the proof of the lemma to Appendix D.

Algorithm 2 describes our method. We efficiently find each class by iterating over closed sets, in decreasing order of size. We start with the full set $[d]$. For each closed set $\overline{S}$ of size $\ell$, we compute its basis $\underline{S}$. By Lemma 3.1, $\overline{S} \setminus \{j\}$ is closed for all $j \in \underline{S}$. We then add each of these closed sets, and further explore them when we reach size $\ell - 1$.

We can see that Algorithm 2 correctly returns all closed sets by an inductive argument. Suppose that we have identified all closed sets of size $\ell$; the base case is trivial since there is only one set, and it must be closed. By Lemma 3.1, for every closed set $\overline{S}$ of size $\ell - 1$, there is a closed set $\overline{S} \cup \{j\}$ for some $j$ in the basis of $\overline{S} \cup \{j\}$. By the inductive assumption, we must have identified this set, and also found $\overline{S}$ by removing $j$ from $\overline{S} \cup \{j\}$. It follows that every closed set of size $\ell - 1$ gets generated by some closed set of size $\ell$.

Algorithm 2 runs in $O(r(d + e))$ time: for each closure—there is one closure for each of the $r$ classes—the algorithm runs a graph exploration in time $d + e$, and then adds at most $d$ closed sets of size $\ell - 1$ to explore.

**Simple Class Optimization** Sometimes running Algorithm 1 as a subroutine in Algorithm 2 can be avoided. For a closed set $\overline{S}$ that is *simple*—$\overline{S}$ is both its own closure and basis—all of its subsets are also simple. To see why, observe that all nodes $j \in \overline{S}$ have a directed path to $Y$ that does not intersect any other node in $\overline{S}$, and hence any other node in a

---

**Algorithm 2** `AllClasses`

---

**Input:** Number of elements $d$, graph $G$
**Output:** All closed sets $\mathcal{C}$
$\mathcal{C}_0 \leftarrow \ldots \leftarrow \mathcal{C}_d \leftarrow \emptyset$      ▷ Closed sets of each size
$\mathcal{C}_d \leftarrow \{\{1, \ldots, d\}\}$      ▷ Only closed set of size $d$
**for** $\ell = d, \ldots, 1$ **do**
     **for** $\overline{S} \in \mathcal{C}_\ell$ **do**
         $\underline{S}, \overline{S} \leftarrow$ `FindClass`$(\overline{S}, G)$
         ▷ Use Lemma 3.1
         **for** $j \in \underline{S}$ **do**
             Add $\overline{S} \setminus \{j\}$ to $\mathcal{C}_{\ell-1}$
**return** $\mathcal{C}_0 \cup \ldots \cup \mathcal{C}_d$

---

subset of $\overline{S}$. It follows that all subsets are irreducible; with Lemma 3.1, we have that all such subsets are also closed. For a simple set, we add an optimization to Algorithm 2 in our implementation so that all the bases of its candidates are cached, avoiding the $O(d + e)$ cost of Algorithm 1 for all subsets of a simple set.

Together, Equation 4 and Algorithm 2 can compute Shapley values in time linear in $r$.

**Proposition 3.2.** *Shapley values of the intervention value function can be exactly computed in $O(r(d + e + T))$ time, where $T$ is the time to evaluate the game $\nu$ on a given coalition, and $r$ is the number of irreducible sets.*

## 4. Approximating do-Shapley Values

While Algorithm 2 allows for exact computation in $O(r(d + e))$ time, the number of irreducible sets $r$ is unknown *a priori*. In resource constrained settings where $r$ may be too large, we would like an approximation technique that operates within a fixed computational budget of $m$ value function queries.

Standard value-function-agnostic estimators sample coalitions from a fixed distribution. While caching can prevent re-evaluation of the SCM for known classes, these estimators suffer from sample redundancy: they blindly generate coalitions that may belong to equivalence classes already queried. As a result, a budget of $m$ queries often yields far fewer than $m$ distinct class values, wasting computational resources on redundant parts of the lattice.

To address this, we seek an estimator that guarantees the discovery of $\min(m, r)$ *distinct* equivalence classes. If $m \geq r$, the method should naturally recover the exact Shapley values to machine precision. If $m < r$, it should prioritize classes with weight to minimize estimation error.

Recall that the Shapley value can be expressed as a weighted sum over equivalence classes:

$$\phi_i = \sum_{j=1}^{r} \nu(c_j) \cdot w_i(c_j). \tag{6}$$

Directly estimating this sum presents a challenge: we cannot sample classes proportional to their weights $w_i(c_j)$ because the classes (and thus their weights) are unknown without graph exploration. Furthermore, a stratified sampling approach—sampling coalitions without replacement to ensure unique classes—proves computationally expensive. As detailed in Appendix F, such a stratified sampling method incurs a cost quadratic in $m$, which defeats the purpose of fast approximation.

**Boundary Sampling**  We propose a *boundary sampler*, a targeted graph exploration strategy that achieves sample efficiency in time linear in $m$. Instead of sampling blindly from the powerset, we maintain candidate classes adjacent to those we have already visited.

The algorithm, described in Algorithm 3, proceeds by maintaining a priority queue of candidate classes, ordered by their expected weight magnitude $\mathbb{E}_i[|w_i(c)|]$. In each iteration, we sample a class $c$ proportional to the expected magnitude of its weight, evaluate $\nu(c)$, and add it to our sampled set. We then expand the candidate by generating the neighbors of $c$ in the lattice. Specifically, for a class $c$ with basis $\underline{S}$ and closure $\overline{S}$, the neighbors are defined as either lower neighbors $\{\overline{S} \setminus \{j\} \mid j \in \underline{S}\}$ or upper neighbors $\{\overline{S} \cup \{j\} \mid j \notin \overline{S}\}$. For each neighbor, we run `FindClass` to determine its canonical representation and weight, adding it to the queue if it has not been seen. This ensures that every query results in a new class. By starting at each level in the lattice, and adding neighbors above and below, we explore the lattice in a balanced way.

We state Algorithm 3, and prove the following upper bound on its runtime, in Appendix D.

**Proposition 4.1.** *Algorithm 4 runs in $O(m \cdot d(d + e))$ time, where $m$ is the query budget, $d$ is the number of features, and $e$ is the number of edges in the graph.*

**Estimation via Simulation**  After running the boundary sampler, we utilize the set of sampled classes $\mathcal{C}$ to compute the Shapley values, as shown in Algorithm 3.

*Case $m \geq r$:* If the queue empties before the budget is reached, we have identified all irreducible sets. We proceed to compute the exact Shapley values using Equation 6.

*Case $m < r$:* We leverage the fact that we have paid the cost to evaluate $\nu(c)$ for all $c \in \mathcal{C}$. We construct a *simulated* estimator, described in Appendix E. This allows us to generate many samples at no additional query cost, reducing the variance of the estimator while balancing time complexity.

We run Algorithm 3 where the base estimators are the current state-of-the-art value-function-agnostic Shapley value estimators `LeverageSHAP` (Musco & Witter, 2025) and `RegressionMSR` (Witter et al., 2025).

## 5. Identifiability

Whenever a causal query $P_S(T)$ is uniquely determined by the graph and dataset, we say the query is (non-parametrically) *identifiable*. However, this is not always the case; we include an illustrative example in Appendix G.

Computing the do-Shapley value requires verifying the identifiability of component queries $\nu(S)$ via the ID algorithm (Shpitser & Pearl, 2006). Even with class-based grouping, this necessitates $r$ separate tests. Conducting these checks sequentially during estimation is risky, as a late discovery of

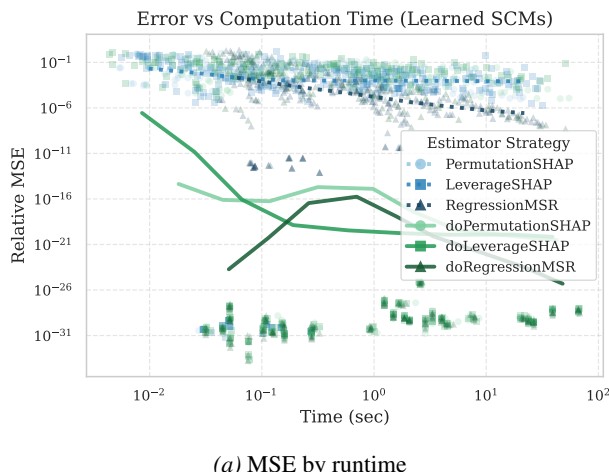

*(a)* MSE by runtime

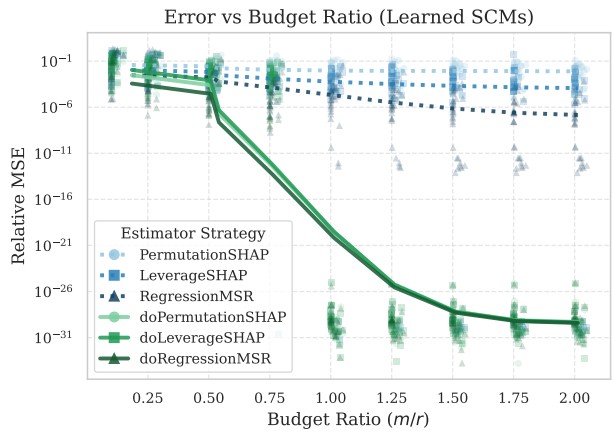

*(b)* MSE by budget ratio

*Figure 5.* **Estimator Convergence on Learned SCMs.** (a) Comparison of relative MSE by runtime, with median lines. The doEstimator variants give much lower error even on a time-adjusted comparison with SOTA Shapley value estimators employing the FRA caching of Parafita et al. (2025). (b) Comparison of relative MSE by budget ratio $m/r$. The doEstimators achieve machine precision when $m \geq r$, and start substantially outperforming the SOTA estimators as $m$ approaches $r$.

---

**Algorithm 3** `doEstimator`

**Input:** Budget $m$, Game $\nu$, `BaseEstimator`, Sampling multiplier $k$
**Output:** Shapley values estimates $\phi \in \mathbb{R}^d$
▷ Step 1: Sampling Phase
$\mathcal{C}$, `allSampled` $\leftarrow$ `BoundarySampler`$(m, G)$
Query $\nu(c)$ for all $c \in \mathcal{C}$          ▷ $\min(r, m)$ queries
▷ Step 2A: Exact Computation
**if** `allSampled` **then**
    $\phi \leftarrow \mathbf{0}$
    **for** each class $c \in \mathcal{C}$ **do**
        $\phi_i \leftarrow \phi_i + \nu(c) \cdot w_i(c)$ **for all** $i$
    **return** $\phi$
▷ Step 2B: Run `BaseEstimator`
$\mathcal{D} \leftarrow$ `SimulatedSampler`$(\mathcal{C}, k \cdot m)$ ▷ Algorithm 5
**return** `BaseEstimator`$(\mathcal{D})$

---

non-identifiability renders all prior computation wasted. To eliminate this overhead, we present a theorem establishing a linear-time check for global identifiability.

**Theorem 5.1.** *The do-Shapley value $\phi_i$ is non-parametrically identifiable if, and only if, $\forall j \in [d]$, $\nu(\{j\})$ is non-parametrically identifiable.*

The necessary background knowledge and proof is left for Appendix G.

Consequently, we can execute the ID algorithm on the $d$ singleton coalition queries and, if all are identifiable, the do-Shapley value will be identifiable and we can proceed with its estimation. If not, further parametric assumptions, or the inclusion of instrumental variables, will be required to

ensure identifiability. Regardless, this result prevents practitioners from estimating do-Shapley values only to learn the outcome was not identifiable to begin with. Additionally, it reduces the number of calls to the ID algorithm from $r$ to $d$.

## 6. Experiments

We evaluate the performance of our exact algorithm and boundary sampling estimators on a diverse set of real-world datasets. Our experiments are designed to investigate three key questions: (1) To what extent does the number of irreducible sets $r$ reduce the complexity compared to the worst-case $2^d$ in real-world dependencies? (2) Do our structure-aware estimators outperform state-of-the-art model-agnostic baselines under fixed query budgets? (3) How does the learned causal structure qualitatively influence feature attribution error?

**Data.** We utilize the TALENT benchmark (Liu et al., 2025), a large-scale repository of tabular datasets. To enable reliable evaluation against exhaustive baselines, we restrict to datasets whose *post-pruning* dimension (after restricting to the ancestors of the target $Y$) permits exact computation of $\nu(S)$ over all coalitions $S \subseteq [d]$ within our computational budget. For each dataset, we select a set of test instances $\mathbf{x}$ and report errors aggregated across instances and datasets. Additional dataset-level details (including the resulting dimensions after pruning) are provided in the appendix.

**SCM Generation.** Since real-world datasets lack ground-truth causal graphs, we learn SCMs from data to serve as the ground-truth games $\nu$. For each dataset, we employ the Greedy Relaxed Search Procedure (GRaSP) (Lam et al., 2022) with a BIC score to learn a Completed Partially Di-

rected Acyclic Graph (CPDAG), which is converted to a DAG greedily. Not all features in a dataset causally influence the target. Following the definition of do-Shapley, we prune the learned graph to the ancestral set of the target variable $Y$; nodes with no directed path to $Y$ have a null Shapley value and are removed. We then fit non-linear structural equations using Gradient Boosting Regressors (Friedman, 2001) to model the conditional distributions $\mathbb{E}[X_i \mid \text{Pa}(X_i)]$. This learned SCM acts as our oracle $\nu(S)$.

**Methods.** We evaluate our framework against two state-of-the-art model-agnostic estimators: `RegressionMSR` (Witter et al., 2025) and `LeverageSHAP` (Musco & Witter, 2025). These serve as our structure-agnostic baselines, estimating Shapley values by directly sampling and querying coalitions from the full powerset, while still employing the FRA caching of Parafita et al. (2025). To explicitly isolate the gains attributable to our graph-theoretic insights, our proposed estimators `doRegressionMSR` and `doLeverageSHAP` are not fundamentally new regression techniques. Rather, they repurpose the *same* estimation machinery used in the baselines. The main difference lies in the data generation process: whereas the baselines query random coalitions (with FRA-caching), our methods train the estimators on the distinct equivalence classes recovered by the boundary sampler (as described in Algorithm 3). This allows us to compare "structure-aware" versus "structure-agnostic" sampling while holding the estimation logic constant.

**Lattice Complexity Reduction**  The efficiency of our exact algorithm relies on $r \ll 2^d$. In Figure 4, we plot the number of irreducible sets $r$ against the dimension $d$ for 156 datasets from the TALENT benchmark. We observe that for real-world data, the number of irreducible sets $r$ often remains significantly below $2^d$.

**Estimation Efficiency**  We evaluate estimation error by query budget $m$, defined relative to the number of irreducible sets $r$. Figure 5 presents Relative MSE for all estimators on learned SCMs, plotted against runtime and budget ratio $m/r$. A similar plot for the posited SCMs of Ankan & Textor (2024) is included in Appendix H.

In the sparse budget regime where we cannot fully explore the lattice, `doRegressionMSR` (blue) demonstrates superior sample efficiency, consistently achieving the lowest error. This advantage becomes increasingly pronounced as the budget approaches $r$. By prioritizing the discovery of distinct equivalence classes via boundary sampling, our method minimizes redundant queries that plague standard samplers.

A phase transition occurs at $m = r$ (indicated by the red vertical line). Once the budget allows for full lattice exploration, our boundary sampler identifies that all irreducible sets have

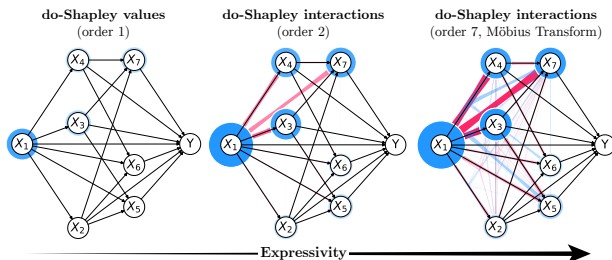

**Figure 6.** Interactions offer an expressive explanation framework.

been found. At this point, the algorithm switches to the exact computation described in Section 3. Consequently, the MSE for both `doRegressionMSR` and `doLeverageSHAP` drops precipitously to machine precision. In contrast, the structure-agnostic baselines (green and purple) continue to sample coalitions with replacement, exhibiting a slow convergence rate and failing to achieve exactness even with double the necessary budget ($m = 2r$).

## 7. Generalizations

While it provides a principled framework for attributing outcomes to individuals variables, the Shapley values are only one way of formalizaing attribution. Fortunately, our work generalizes beyond Shapley values in two important ways:

**a)** Our techniques extend to *probabilistic values* where the weighting function $p$ is specified by other distributions, e.g., Banzhaf values, weighted Banzhaf values, or Beta Shapley values.

**b)** Interaction attribution measure the effect of changes in the value function with respect to subsets $T$ rather than singletons $i$, as shown in Figure 6. Our techniques extend to these interaction attributions; please see Appendix I for details.

## 8. Conclusion

We address the computational challenges of computing do-Shapley values by reformulating them in terms of the structural equivalence classes induced by the underlying SCM. By traversing the lattice of closed sets, our exact algorithm scales with the number of irreducible sets $r$ rather than the worst-case $2^d$, and our boundary sampling estimators extend this advantage to fixed-budget settings, recovering exact values once the query budget satisfies $m \geq r$. We further show that verifying non-parametric identifiability of the full do-Shapley value reduces to a linear $O(d)$ check over the singleton interventions.

Two limitations temper these gains. First, although real-world causal graphs are typically sparse enough that $r \ll 2^d$,

this is not guaranteed: in the worst case $r$ grows as $2^d$, and our methods inherit exponential complexity on such graphs. Deriving tighter topology-dependent bounds on $r$—beyond the strict bounds $2^p + (d - p) \leq r \leq 2^d - 2^{d-p} + 1$, where $p = |Pa_Y|$—remains an open problem. Second, and more fundamentally, our methods return the do-Shapley values of the value function $\nu$ *as specified*. Like causal attribution methods generally, they rely on a correctly specified SCM: if the graph structure is misspecified, the resulting attributions faithfully describe the flawed model rather than the true data-generating process. Characterizing the sensitivity of do-Shapley values to graph misspecification is a natural direction for future work.

Despite these caveats, our framework makes causal explanation substantially more practical, turning an exponential computation into one governed by the actual structure of the problem. Moreover, the central idea—partitioning the powerset via irreducible sets—is not specific to the Shapley value: it applies directly to the broader family of probabilistic values, including Banzhaf and Beta Shapley values, as well as to interaction indices. We hope this structural perspective encourages further work at the intersection of causal inference and game-theoretic attribution.

## Impact Statement

This paper presents work whose goal is to advance the field of machine learning. There are many potential societal consequences of our work, none of which we feel must be specifically highlighted here.

## Acknowledgment

Álvaro Parafita received funding from the AI4Science fellowship within the "Generacion D" initiative, Red.es, Ministerio para la Transformación Digital y de la Función Pública, for talent attraction (C005/24-ED CV1), funded by the European Union NextGenerationEU funds, through PRTR. Axel Brando received funding from the Horizon Europe Programme under the AI4DEBUNK Project (https://www.ai4debunk.eu), grant agreement num. 101135757. Maximilian Muschalik gratefully acknowledges funding by the Deutsche Forschungsgemeinschaft (DFG, German Research Foundation): TRR 318/3 2026 – 438445824. Tomàs Garriga gratefully acknowledges Novartis for sponsoring his industrial PhD, and the Government of Catalonia's Industrial PhDs Plan for funding part of his research.

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

# A. Notation Summary Table

*Table 1.* Summary of Key Notation

| Symbol | Description |
|---|---|
| $d$ | Number of features/variables |
| $Y$ | Target outcome variable |
| $G$ | Causal graph (Directed Acyclic Graph or Acyclic Directed Mixed Graph) |
| $\mathbf{x}$ | Specific observation/instance of interest |
| $S, T, U$ | Coalitions (subsets of features), e.g., $S \subseteq [d]$ |
| $\nu(S)$ | Interventional value function, $\mathbb{E}[Y \mid \mathrm{do}(S = \mathbf{x}_S)]$ |
| $\phi_i$ | do-Shapley value for feature $i$ |
| $p_{|S|}$ | Shapley weight for a coalition of size $|S|$ |
| $\underline{S}$ | Basis of $S$ (smallest coalition in the equivalence class) |
| $\overline{S}$ | Closure of $S$ (largest coalition in the equivalence class) |
| $c_j$ | $j$-th equivalence class of coalitions |
| $r$ | Total number of equivalence classes (irreducible sets) |
| $w_i(c)$ | Class weight of feature $i$ in equivalence class $c$ |
| $m$ | Fixed computational budget (number of value function queries) |
| $Pa_G(X)$ | Set of parents of node $X$ in graph $G$ |
| $An_G(X)$ | Set of directed ancestors of node/set $X$ in graph $G$ |
| $De_G(X)$ | Set of directed descendants of node/set $X$ in graph $G$ |
| $G_{\overline{S}}$ | Intervened graph with incoming edges to nodes in $S$ removed |
| $\mathcal{C}(G)$ | Set of maximal C-components of graph $G$ |
| $R$ | Root set of an ADMG |

# B. Additional Related Work

In many practical scenarios, underlying models act as black boxes, precluding exact Shapley value analysis and necessitating the use of model-agnostic estimators (Chen et al., 2023; Muschalik et al., 2024b). Practitioners typically rely on Monte Carlo sampling (Castro et al., 2009; Fumagalli et al., 2023; Wang & Jia, 2023; Kolpaczki et al., 2024a;b) or regression-based approximations like `KernelSHAP` (Lundberg & Lee, 2017). The latter has seen continual refinement, recently expanding to support feature interactions (Tsai et al., 2023; Fumagalli et al., 2024) and leverage score enhancements (Musco & Witter, 2025). Conversely, exact computation of Shapley values becomes feasible only when a model's architecture possesses an exploitable structure (Rozemberczki et al., 2022). This structural advantage has driven the development of highly efficient, model-specific algorithms. Prominent examples include tree-based methods such as TreeSHAP (Lundberg et al., 2020; Yu et al., 2022), its interventional adaptations (Zern et al., 2023b), and interaction variants like TreeSHAP-IQ (Muschalik et al., 2024c). Similar exact strategies have been successfully derived for a variety of other architectures, ranging from linear models (Štrumbelj & Kononenko, 2014) and K-nearest neighbor data valuation (Jia et al., 2019; Wang et al., 2023; 2024) to product-kernel networks (Mohammadi et al., 2025a), Gaussian processes (Mohammadi et al., 2025b), and graph neural networks (Muschalik et al., 2025). Finally, modern research is bridging the gap between *estimation* and *exact computation* through surrogate modeling (Butler et al., 2025), demonstrating that fitting auxiliary models to the value function enables the efficient and precise extraction of Shapley values (Witter et al., 2025).

## C. Background on Structural Causal Models

A **Causal graph** is usually described as a Directed Acyclic Graph (DAG) $G$, where every node represents a measured random variable and every directed edge represents a relationship cause $\rightarrow$ effect. These graphs often include dashed bidirected edges between pairs of nodes ($X \leftrightarrow Y$) as a shorthand for the existence of an unobserved latent variable $U$ that acts as a confounder between them ($X \leftarrow U \rightarrow Y$). Additionally, it is assumed that every measured node $X$ has a latent exogenous noise variable, denoted $E_X$, with the associated edge $E_X \rightarrow X$.

A **Structural Causal Model** (SCM) is a probabilistic model based on such a causal graph $G$ with a probability distribution for all latent nodes (i.e., all confounders and $E_X$) and, for each measured node $X$, functions $f_X$ such that $X := f_X(Pa_G(X))$, taking the values of all parents of $X$ in $G$ including latent variables. From this model, a probability distribution over the measured variables $V$ emerges, $P(V)$, as well as the intervened model $M_{do(S=s)}$, where the $do$ operator conveys an intervention on all nodes $X \in S$ replacing their functions $f_X$ by the assignment $x := s_X$. This effectively removes all incoming edges to the intervened nodes from the graph, and results in a new probability distribution for the intervened model, $P(V \mid do(S = s))$, also denoted $P_s(V)$ or, for arbitrary intervention values, $P_S(V)$.

Given a dataset and its assumed underlying causal structure, we can train an SCM following that graph to learn the distribution of the dataset. Afterwards, one can employ procedures on the SCM to estimate **causal queries** of the form $P_X(Y)$. Additionally, if the causal query is non-parametrically *identifiable* (more details in Section 5), these estimations resulting from the SCM are necessarily equivalent to what the true data generating process would return if we had access to it. Therefore, we can employ these trained SCMs to estimate the do-SHAP value functions $\nu(S)$ as long as they are identifiable. For more details about this approach, please refer to (Parafita & Vitrià, 2022).

# D. Delayed Proofs

Let $\underline{S}$ and $\overline{S}$ be the basis and closure of a coalition $S$, respectively. By definition, $\underline{S} \subseteq S \subseteq \overline{S}$, $\overline{\overline{S}} = \overline{S}$ and $\underline{\underline{S}} = \underline{S}$.

Let us now prove an essential result that explains the soundness of Algorithm 2.

**Lemma 3.1** *Let $\overline{S} \subset [d]$ be a closed set with basis $\underline{S}$. Then*

1. *For all $j \in \underline{S}$, $\overline{S} \setminus \{j\}$ is closed.*

2. *If $\overline{S} \neq [d]$, there exists $j \in [d] \setminus \overline{S}$ so that $\overline{S} \cup \{j\}$ is closed and $j$ is in the basis of $\overline{S} \cup \{j\}$.*

*Proof of Lemma 3.1.* We will first show that $\overline{S} \setminus \{j\}$ is closed for all $j \in \underline{S}$. To do so, it suffices to show that, for all nodes *not* in $\overline{S} \setminus \{j\}$, there is a directed path to $Y$ that does not intersect $\overline{S} \setminus \{j\}$. This is clearly true for all nodes not in $\overline{S}$ since $\overline{S}$ is itself closed. It must also be true for $j$ since $j \in \underline{S}$, i.e., there is a directed path from $j$ to $Y$ that does not intersect $S$. The first statement follows.

Next, we will show that for $S \neq [d]$, there exists $j \in [d] \setminus \overline{S}$ so that $\overline{S} \cup \{j\}$ is closed. Let $T = [d] \setminus \overline{S}$. Since $G$ is a finite DAG, there is a topological order on the nodes. Consider a node $j \in T$ with no ancestors in $T$, either because they are all in $\overline{S}$, or it has no ancestors. To prove that $\overline{S} \cup \{j\}$ is closed, it suffices to show that, for all nodes *not* in $\overline{S} \cup \{j\}$, there is a directed path to $Y$ that does not intersect $\overline{S} \cup \{j\}$. Since $\overline{S}$ is closed, all of the nodes in $T \setminus \{j\}$ have paths to $Y$ that do not intersect $\overline{S}$. Note that $j$ cannot be in any of these paths, since $j$ has no ancestors in $T$, hence all nodes in $T \setminus \{j\}$ must actually have paths that do not intersect $\overline{S}$ or $j$. Furthermore, since $j \in T$, it must have a directed path to $Y$ that does not intersect $\overline{S}$. Therefore, $j$ must be in the basis of $\overline{S} \cup \{j\}$. The second statement follows. $\square$

**Proposition 4.1**. *Algorithm 4 runs in $O(m \cdot d(d + e))$ time, where $m$ is the query budget, $d$ is the number of features, and $e$ is the number of edges in the graph.*

*Proof of Proposition 4.1.* The algorithm performs exactly $m$ iterations of the main `while` loop. In each iteration, we process one class $c$. The cost of processing a class is dominated by generating its neighbors and invoking `FindClass` for each. A class with closure $\overline{S}$ and basis $\underline{S}$ has $|\underline{S}|$ lower neighbors and $d - |\overline{S}|$ upper neighbors; thus, the total number of neighbors is bounded by $d$. For each neighbor, we execute `FindClass`, which requires a graph traversal taking $O(d + e)$ time. Therefore, the work per iteration is $O(d(d + e))$. Over $m$ iterations, the total time complexity is $O(m \cdot d(d + e))$. This is linear in the budget $m$, ensuring the method is scalable for anytime estimation. $\square$

For completeness, we now prove the following simple statements about the irreducible set structure.

**Lemma D.1.** *For all $j \in \overline{S}$, every directed path $p$ from $j$ to $Y$ intersects $\underline{S}$.*

*Proof.* If $j \in \underline{S}$, it is trivial. Assume $j \in \overline{S} \setminus \underline{S}$. Consequently, $\exists k \in p \cap (S \setminus \{j\})$. If $k \notin \underline{S}$, since $k \in S \setminus \{j\} \subseteq \overline{S}$, we can repeat the argument to find a new $k' \in S \setminus \{j, k\}$. Since the graph is acyclic and $S$ is finite, eventually we will reach a $k \in \underline{S}$, which proves the lemma. $\square$

**Lemma D.2.** *The closure of the basis of $S$ is the closure of $S$: $\overline{(\underline{S})} = \overline{S}$.*

*Proof.* By Lemma D.1, all nodes in $\overline{S}$ must have all their directed paths to $Y$ intersected with $\underline{S}$, so they belong in $\overline{(\underline{S})}$. In the same way, all nodes in $\overline{(\underline{S})}$ must have all directed paths to $Y$ intersecting with $\underline{S} = \underline{\underline{S}} \subseteq S$, so they belong in $\overline{S}$. $\square$

**Lemma D.3.** *The basis of the closure of $S$ is the basis of $S$: $\underline{(\overline{S})} = \underline{S}$.*

*Proof.* Let us prove that $\underline{(\overline{S})} \subseteq \underline{S}$. Assume by absurd that a node $j \in \underline{(\overline{S})} \subseteq \overline{S}$ is not in $\underline{S}$; therefore, $j \in \overline{S} \setminus \underline{S}$. By Lemma D.1, all directed paths $p$ from $j$ to $Y$ intersect with $\underline{S} \setminus \{j\}$, but since $j \in \underline{(\overline{S})}$, there is also a path that does not intersect with $\overline{S} \setminus \{j\}$, therefore neither by $\underline{S} \setminus \{j\}$, contradicting the assumption. Necessarily, $j \in \underline{S}$ and $\underline{(\overline{S})} \subseteq \underline{S}$.

Now, to prove $\underline{S} \subseteq \underline{(\overline{S})}$, let us proceed by absurd. Assume $\exists j \in \underline{S} \setminus \underline{(\overline{S})}$. Since $j \in \underline{S} \subseteq \overline{S}$, $j \in \overline{S} \setminus \underline{(\overline{S})}$, so all directed paths from $j$ to $Y$ must intersect $\underline{(\overline{S})} \setminus \{j\}$ by Lemma D.1. Since $\underline{(\overline{S})} \subseteq \underline{S}$, $\underline{S} \setminus \{j\}$ must also intersect all paths, so $j \notin \underline{S}$, contradicting the assumption. $\square$

---

**Algorithm 4** `BoundarySampler`

---

**Input:** Budget $m$, Graph $G$
**Output:** Sampled classes $\mathcal{C}$, flag `allSampled`
$\mathcal{C} \leftarrow \emptyset$        ▷ Sampled classes
$\mathcal{Q} \leftarrow \emptyset$        ▷ Queue mapping classes to weights
$\mathcal{C}_{\text{seen}} \leftarrow \emptyset$
▷ Helper to calculate weight and enqueue
**function** `Enqueue`$(S)$
   $c \leftarrow$ `FindClass`$(S, G)$        ▷ Get basis and closure
   **if** $c \in \mathcal{C}_{\text{seen}}$ **then**
      **return**
   Add $c$ to $\mathcal{C}_{\text{seen}}$
   $\mathcal{Q}[c] \leftarrow \mathbb{E}_i[|w_i(c)|] + \epsilon$        ▷ Add $\epsilon$ to ensure valid dist.
▷ Phase 1: Warm-start
**for** $\ell = 1$ **to** $d$ **do**
   Sample random set $S \subset [d]$ where $|S| = \ell$
   `Enqueue`$(S)$
▷ Phase 2: Weighted Graph Traversal
**while** $|\mathcal{C}| < m$ **and** $\mathcal{Q} \neq \emptyset$ **do**
   Sample $c$ from $\mathcal{Q}$ with prob $\propto \mathcal{Q}[c]$
   Remove $c$ from $\mathcal{Q}$ and add to $\mathcal{C}$
   $\underline{S}, \overline{S} \leftarrow$ basis and closure of $c$
   $N_{\text{below}} \leftarrow \{\overline{S} \setminus \{j\} \mid j \in \underline{S}\}$
   $N_{\text{above}} \leftarrow \{\overline{S} \cup \{j\} \mid j \notin \overline{S}\}$
   **for** candidate set $S' \in N_{\text{below}} \cup N_{\text{above}}$ **do**
      `Enqueue`$(S')$
**return** $\mathcal{C}, \mathcal{Q} == \emptyset$

---

**Definition D.4** (Equivalent Coalitions). Let us define $S \sim T$ when $\underline{S} = \underline{T}$, which is an equivalence relationship (trivially reflexive, symmetric and transitive due to the basis being unique).

**Proposition D.5.** *The equivalence class $[S]$ is given by any coalition between its basis and closure:* $[S] = \{T \mid \underline{S} \subseteq T \subseteq \overline{S}\}$.

*Proof.* By Lemma D.2, if $\underline{T} = \underline{S}, \overline{T} = \overline{(\underline{T})} = \overline{(\underline{S})} = \overline{S}$, then $\underline{S} = \underline{T} \subseteq T \subseteq \overline{T} = \overline{S}$.

Now note that the closure is monotonous w.r.t. inclusion: if $S \subseteq T, \forall j \in \overline{S}$ we know that all directed paths from $j$ to $Y$ are intersected by $S$, therefore also by $T \supseteq S$, so $j \in \overline{T}$ and $\overline{S} \subseteq \overline{T}$. With this result, if we assume $\underline{S} \subseteq T \subseteq \overline{S}$, $\underline{S} \subseteq T$ implies $\overline{S} = \overline{(\underline{S})} \subseteq \overline{T}$ and $T \subseteq \overline{S}$ implies $\overline{T} \subseteq \overline{\overline{S}} = \overline{S}$. Therefore, $\overline{T} = \overline{S}$. Finally, applying Lemma D.3, $\underline{T} = \underline{(\overline{T})} = \underline{(\overline{S})} = \underline{S}$, which proves that $S \sim T$. $\qquad\square$

**Proposition D.6.** *Given an equivalence class $[S]$, $\nu$ has identical value for all its elements:* $\forall T \in [S], \nu(\underline{S}) = \nu(T) = \nu(\overline{S})$.

*Proof.* By definition, all nodes in $\overline{S} \setminus \underline{S}$ are *blocked* from reaching $Y$ by $\underline{S}$ through front-door paths. Back-door paths, if they exist, are removed by the intervention on themselves. Therefore, by the third rule of do-Calculus (Pearl, 2009), all of these nodes can be removed from the intervention set, rendering $\nu(\underline{S}) = \nu(T) = \nu(\overline{S})$. $\qquad\square$

# E. Simulated Sampling from Irreducible Sets

In this appendix, we detail the `SimulatedSampler`, the engine behind Step 2B of the `doEstimator`. This component allows us to sample coalitions efficiently from the specific sub-lattice defined by the discovered equivalence classes.

## E.1. Algorithm and Explanation

The challenge in simulating samples from the discovered classes is that the union of these classes does not form a simple structure (like a full powerset). A naive rejection sampling approach—sampling from the full powerset and keeping only those in $\mathcal{C}$—would be inefficient if the volume of $\mathcal{C}$ is small relative to $2^d$.

Instead, our sampler (Algorithm 5) adopts a constructive approach:

- Normalization: We first calculate the total "volume" of available coalitions within the known classes. For each class $c$ with basis $\underline{S}$ and closure $\overline{S}$, the number of subsets of size $s$ contained in $c$ is given by $\binom{|\overline{S}| - |\underline{S}|}{s - |\underline{S}|}$.

- Scale Calibration: Standard Shapley weights are defined for the entire powerset. To sample validly from our restricted support, we solve for a scaling factor $\gamma$ such that the expected number of samples drawn matches our target budget $B_{\text{sim}}$. This is achieved via binary search (Lines 7-15).

- Stratified Generation: We iterate through each class $c \in \mathcal{C}$ and each valid subset size $s$. For each size, we compute the expected number of samples $N_{c,s}$. We use probabilistic rounding to convert this expectation into an integer count, and then generate that many unique subsets from class $c$ using a combinatorial number system (Lines 22-26).

---

**Algorithm 5** `SimulatedSampler`

---

1: **Input:** Discovered classes $\mathcal{C}$, Simulation Budget $B_{\text{sim}}$, Weights $w$ (per size)
2: **Output:** Dataset of coalitions $\mathcal{D} = \{(S_k, \nu(c_k), p_k)\}$
3: $\mathcal{D} \leftarrow \emptyset$
4: ▷ Step 1: Count available coalitions per size across all classes
5: $N_{\text{avail}}[s] \leftarrow 0$ for $s \in 0 \dots d$
6: **for** each class $c \in \mathcal{C}$ **do**
7:    Let $n_{\text{free}} = |\overline{S}| - |\underline{S}|$
8:    **for** $j = 0$ **to** $n_{\text{free}}$ **do**
9:        $N_{\text{avail}}[|\underline{S}| + j] \leftarrow N_{\text{avail}}[|\underline{S}| + j] + \binom{n_{\text{free}}}{j}$
10: ▷ Step 2: Calibrate sampling scale $\gamma$
11: Define $E[\text{samples}](\gamma) = \sum_{s=0}^{d} N_{\text{avail}}[s] \cdot \min\left(\gamma \frac{w_s}{\binom{d}{s}}, 1\right)$
12: Find $\gamma^*$ via binary search such that $E[\text{samples}](\gamma^*) \approx B_{\text{sim}}$
13: ▷ Step 3: Constructive Sampling
14: **for** each class $c \in \mathcal{C}$ **do**
15:    Let $I_{\text{free}}$ be indices in $\overline{S} \setminus \underline{S}$
16:    **for** $j = 0$ **to** $|I_{\text{free}}|$ **do**
17:        Size $s \leftarrow |\underline{S}| + j$
18:        Prob $p \leftarrow \min\left(\gamma^* \frac{w_s}{\binom{d}{s}}, 1\right)$
19:        Count $K \leftarrow \binom{|I_{\text{free}}|}{j}$
20:        Expected count $\mu \leftarrow K \cdot p$
21:        $N_{\text{draw}} \leftarrow \lfloor \mu \rfloor + \text{Bernoulli}(\mu - \lfloor \mu \rfloor)$          ▷ Probabilistic rounding
22:        **if** $N_{\text{draw}} > 0$ **then**
23:            Generate $N_{\text{draw}}$ unique combinations $C_{\text{sub}} \subseteq I_{\text{free}}$ of size $j$
24:            **for** each combination $\sigma \in C_{\text{sub}}$ **do**
25:                $S \leftarrow \underline{S} \cup \sigma$
26:                Add $(S, \nu(c), p)$ to $\mathcal{D}$
27: **return** $\mathcal{D}$

---

### E.2. Runtime Analysis

**Proposition E.1.** *The* `SimulatedSampler` *generates $N_{sim}$ samples in $O(N_{sim} \cdot d + |\mathcal{C}| \cdot d)$ time.*

*Proof.* The algorithm consists of three main parts:

1. Counting ($O(|\mathcal{C}| \cdot d)$): We iterate over each class once. For each class, we perform a loop over its "free" size range, which is at most $d$. The binomial coefficient calculations can be done in $O(1)$ with pre-computation.

2. Calibration ($O(d \cdot \log(1/\epsilon))$): The binary search evaluates the expected sample sum a constant number of times. Each evaluation sums over $d$ sizes.

3. Generation ($O(N_{\text{sim}} \cdot d)$): The outer loops iterate over classes and sizes, but the inner generation logic (Lines 24-28) executes exactly $N_{\text{sim}}$ times in total (by definition of the calibrated budget). Generating a combination of size $k$ using the combinatorial number system or direct sampling takes $O(d)$. Thus, the generation phase scales linearly with the number of output samples.

Dominating terms yield a total complexity of $O(N_{\text{sim}} \cdot d + |\mathcal{C}| \cdot d)$, which is highly efficient given that no SCM evaluations are performed. $\square$

## F. Stratified Sampling

Ideally, to estimate the Shapley value for player $i$, we would sample each class $c$ proportional to the magnitude of its weight $w_i(c)$. However, the underlying class structure (the mapping of sets to values) is unknown prior to sampling, making the direct computation of $w_i(c)$ impossible.

To address this, we consider an adaptive sampling scheme based on the natural distribution suggested by the Shapley weights. We sample a set $S$ containing $i$ with probability $p_{\ell-1}$, and a set $S$ excluding $i$ with probability $p_\ell$. Note that while these probabilities are derived from the Shapley weights, they do not perfectly correspond to the final importance weights because the contribution of $i$ cancels to zero when $i$ is effectively a "null" player (i.e., in the closure but not in the basis).

The proposed **do-Good estimator** samples according to these distributions *without replacement*. Sampling classes without replacement is crucial for the estimator to achieve exactness when the sampling budget covers the effective support of the game ($m \geq r$). To achieve this, we maintain weighted counts of the remaining "mass" of the distributions for each player $i$. We define the remaining mass for sets including $i$ ($\mu^{(+)}$) and excluding $i$ ($\mu^{(-)}$) as:

$$\mu_{\ell,i}^{(+)} = p_{\ell-1} \sum_{S:|S|=\ell, i \in S} \mathbb{1}[S \text{ not seen}], \tag{7}$$

$$\mu_{\ell,i}^{(-)} = p_\ell \sum_{S:|S|=\ell, i \notin S} \mathbb{1}[S \text{ not seen}]. \tag{8}$$

Initially, before any classes have been sampled, these initialize to the full binomial sums: $\mu_{\ell,i}^{(+)} = p_{\ell-1}\binom{d-1}{\ell-1}$ and $\mu_{\ell,i}^{(-)} = p_\ell\binom{d-1}{\ell}$.

### Sampling Procedure

The sampling process proceeds hierarchically to determine whether to sample proportional to $\mu_{\ell,i}^{+}$ or $\mu_{\ell,i}^{-}$:

1. We first select an index $i$ uniformly from $[d]$.

2. We sample an inclusion indicator $z \in \{-, +\}$, where

$$\Pr(z = +) = \frac{\sum_{\ell=1}^{d} \mu_{\ell,i}^{(+)}}{\sum_{\ell=1}^{d} \mu_{\ell,i}^{(+)} + \sum_{\ell=0}^{d-1} \mu_{\ell,i}^{(-)}}.$$

3. We sample a set size $\ell$ with probability proportional to the remaining mass $\mu_{\ell,i}^{(z)}$.

Once $\ell$ and $z$ are determined, we must sample a specific set $S$ of size $\ell$ (containing $i$ if and only if $z = +$) *uniformly* from the collection of all such sets that belong to currently unseen classes. This step is non-trivial; it depends on the number of valid completions available in the unseen space. As detailed in Algorithm 7, computing these counts requires a linear pass through the history of previously discovered classes.

We analyze the computational complexity of the `ClassSampler` (Algorithm 6). The runtime is dominated by the requirement to sample uniformly from unseen sets, which necessitates checking consistency against all previously discovered classes.

**Proposition F.1** (Stratified Sampling Complexity). *Let $m$ be the sampling budget (number of iterations) and $d$ be the number of elements (dimension). Assuming set operations (union, intersection, subset checks) take $O(d)$ time, the total time complexity of the `ClassSampler` is $O(m^2 d^2)$.*

Because the algorithm is quadratic in $m$, we turn to the *boundary sampling* method described in Section 4.

*Proof of Proposition F.1.* The analysis proceeds by examining the cost of the helper functions from the bottom up.

**1. Cost of `CountSeen`:** This function iterates through the set of seen classes $\mathcal{C}$. In the $k$-th iteration of the main loop, $|\mathcal{C}| \leq k$. Inside the loop, we perform standard set operations (checking $\underline{S} \subseteq S \cup R$, etc.).

$$T_{\text{count}}(k) = O(|\mathcal{C}| \cdot d) = O(k \cdot d). \tag{9}$$

---

**Algorithm 6** `ClassSampler`

---

**Input:** Number of elements $d$, budget $m$, value function $\nu$, graph $G$
**Output:** Set of seen classes $\mathcal{C}$
Initialize seen classes $\mathcal{C} \leftarrow \emptyset$
Set $\mu_{\ell,i}^{(+)} \leftarrow p_{\ell-1}\binom{d-1}{\ell-1}$ for all $\ell \in \{1, \ldots, d\}$ and $i \in [d]$
Set $\mu_{\ell,i}^{(-)} \leftarrow p_\ell\binom{d-1}{\ell}$ for all $\ell \in \{0, \ldots, d-1\}$ and $i \in [d]$
**for** idx $= 1$ **to** $m$ **do**
    ▷ Sample set from unseen classes
    Sample $i \sim \text{Uniform}([d])$
    Sample $z \sim \text{Bernoulli}\left(\frac{\sum_{\ell=1}^{d} \mu_{\ell,i}^{(+)}}{\sum_{\ell=1}^{d} \mu_{\ell,i}^{(+)} + \sum_{\ell=0}^{d-1} \mu_{\ell,i}^{(-)}}\right)$
    Sample $\ell \propto \mu_{\ell,i}^{(z)}$
    $S \leftarrow \{i\}$ **if** $z = 1$ **else** $\emptyset$
    $S \leftarrow \texttt{SampleUnseenBySize}(\ell, S, \mathcal{C})$
    $\underline{S}, \overline{S} \leftarrow \texttt{FindClass}(S, G)$                     ▷ Find class $c$
    Update all $\mu_{\ell,i}^{(+)}$ and $\mu_{\ell,i}^{(-)}$                ▷ Constant time per $\ell$ and $i$
    Add class $c$ to $\mathcal{C}$
**return** $\mathcal{C}$

---

**2. Cost of `SampleUnseenBySize`:** This function constructs a set $S$ of size $\ell$ element-by-element. The `while` loop runs at most $\ell \leq d$ times. In every iteration, it calls `CountSeen` twice to calculate $N_{\text{in}}$ and $N_{\text{out}}$.

$$T_{\text{sample}}(k) = \sum_{j=1}^{\ell} 2 \cdot T_{\text{count}}(k)$$
$$= O(d \cdot (k \cdot d)) = O(kd^2). \tag{10}$$

**3. Total Cost of `ClassSampler`:** The main algorithm runs for $m$ iterations. In iteration $k$, it calls `SampleUnseenBySize`, performs a graph lookup (`FindClass`), and updates weights.

The function `FindClass` runs in $O(d + e)$ time where $e$ is the number of edges in the graph. Since $e \leq d^2$, the dominant cost remains the combinatorial counting step. The update of $\mu$ values takes $O(d^2)$ but is repeated only $m$ times.

Summing over $m$ iterations:

$$T_{\text{total}} = \sum_{k=1}^{m} \left(T_{\text{sample}}(k) + O(d^2)\right)$$
$$= \sum_{k=1}^{m} O(kd^2)$$
$$= O(d^2) \sum_{k=1}^{m} k \approx O(d^2 m^2). \tag{11}$$

Thus, the total complexity is quadratic in both the dimension and the sample budget. $\qquad\square$

---

**Algorithm 7** `SampleUnseenBySize`

---

**Input:** Size $\ell$, Initial set $S$ (i.e., $\{i\}$ or $\emptyset$), seen classes $\mathcal{C}$
**Output:** A set $S$ of size $\ell$ sampled uniformly from unseen sets
$R \leftarrow [d] \setminus S$                                     $\triangleright$ Available candidates
$\mathcal{C}' \leftarrow \{c : |\underline{S}| \leq \ell \leq |\overline{S}|\}$                  $\triangleright$ Relevant classes
**while** $|S| < \ell$ **do**
  Pick $j$ uniformly from $R$
  $\triangleright$ Completions in seen classes with and without $j$
  $N_{\text{in}}, \mathcal{C}'_{\text{in}} \leftarrow \texttt{CountSeen}(\ell, S \cup \{j\}, R \setminus \{j\}, \mathcal{C}')$
  $N_{\text{out}}, \mathcal{C}'_{\text{out}} \leftarrow \texttt{CountSeen}(\ell, S, R \setminus \{j\}, \mathcal{C}')$
  $\triangleright$ Calculate count of *unseen* completions
  $U_{\text{in}} \leftarrow \binom{|R|-1}{\ell-|S|-1} - N_{\text{in}}$
  $U_{\text{out}} \leftarrow \binom{|R|-1}{\ell-|S|} - N_{\text{out}}$
  $\triangleright$ Pick $j$ proportional to unseen completions
  Sample $b \sim \text{Bernoulli}\left(\frac{U_{\text{in}}}{U_{\text{in}}+U_{\text{out}}}\right)$
  $\mathcal{C}' \leftarrow \mathcal{C}'_{\text{out}}$
  **if** $b = 1$ **then**
     $\mathcal{C}' \leftarrow \mathcal{C}'_{\text{in}}$
     $S \leftarrow S \cup \{j\}$
  $R \leftarrow R \setminus \{j\}$
**return** $S$

---

**Algorithm 8** `CountSeen`

---

**Input:** Size $\ell$, current set $S$, remaining set $R$, seen classes $\mathcal{C}$
**Output:** Number of completions $N$, applicable classes $\mathcal{C}'$
$N \leftarrow 0$
$\mathcal{C}' \leftarrow \emptyset$                                      $\triangleright$ Initialize applicable classes
**for** class $c \in \mathcal{C}$ **do**
  Let $\underline{S}$ be the basis and $\overline{S}$ be the closure of $c$
  $\triangleright$ Check if $S$ and $R$ are consistent with class
  **if** $\underline{S} \not\subseteq (S \cup R)$ or $S \not\subseteq \overline{S}$ **then continue**
  Add $c$ to $\mathcal{C}'$                                        $\triangleright$ Applicable class
  $\triangleright$ Calculate available optional elements in this class
  $n_{\text{options}} \leftarrow |R \cap (\overline{S} \setminus \underline{S})|$
  $\triangleright$ Calculate available, optional spaces
  $n_{\text{spaces}} \leftarrow \ell - |S| - |R \cap \underline{S}|$
  $N \leftarrow N + \binom{n_{\text{options}}}{n_{\text{spaces}}}$
**return** $N$

---

# G. Identifiability Criterion

In this section, we operate exclusively with non-parametric identifiability, following (Pearl, 2009) (Definition 3.2.4), where an interventional causal query $Q[M]$, defined on a causal model $M$ with DAG $G = (V \cup U, E)$ with $V$ measured nodes and $U$ latent nodes, is (non-parametrically) identifiable from $G$ if the query can be computed uniquely from any positive probability of the observed variables Markov-relative[3] to the graph. In other words, if two causal models $M_1$ and $M_2$ with positive probability distributions $P_{M_1}$ and $P_{M_2}$ are both Markov-relative to $G$ and observationally identical to each other ($P_{M_1}(V) = P_{M_2}(V)$), then both models must have the same value for the given query. Note that this definition does not include any assumptions about the functional forms of the causal models nor the probability distributions of the unobserved variables. In other words, non-parametric identifiability cannot assume anything other than the shape of the graph $G$ (including latent confounders).

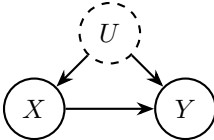

*Figure 7.* Bow-arc graph, with non-identifiable $P_X(Y)$.

**Non-identifiable example.** Consider the Directed Acyclic Graph (DAG) $G = (\{X, Y\} \cup \{U\}, E)$ in Figure 7, where $U$ is an unmeasured variable acting as a latent confounder between $X$ and $Y$. In this graph, $P_X(Y)$ is not identifiable. Let us demonstrate by creating two equivalent positive distributions $P^1$ and $P^2$ Markov-relative to $G$, with different intervened distributions for $P_X(Y)$. We define two latent variables: $U$, the latent confounder between $X$ and $Y$, and $E_Y$, the exogenous noise variable for $Y$, both with the same distribution in both models, $\mathcal{N}(0, 1)$. In terms of their functional assignments, $x := f_X^1(u) = f_X^2(u) = u$, while $y := f_Y^1(x, u, e_y) = x - u + e_y$ and $y := f_Y^2(x, u, e_y) = x \cdot 0 + e_y$. Note that all distributions $P^1(X), P^2(X), P^1(Y \mid X), P^2(Y \mid X)$ are identical, $\mathcal{N}(0, 1)$, and so is $P^1(X, Y) = P^1(X)P^1(Y \mid X) = P^2(X)P^2(Y \mid X) = P^2(X, Y)$, and they are positive. However, $P^1(Y \mid do(X = 0)) \sim N(0, 2)$ while $P^2(Y \mid do(X = 0)) \sim N(0, 1)$. Therefore, $P_X(Y)$ is not identifiable.

Non-parametric identifiability can be demonstrated if we can find an estimand of the causal query only containing observational terms. In particular, the ID algorithm (Shpitser & Pearl, 2006), described in Section G.1 and presented in simplified form in Algorithm 9, determines non-parametric identifiability w.r.t. a causal graph $G$ of any non-conditional, interventional query $P_X(Y)$, for any disjoint subsets of variables $X, Y$.

Note that any coalition value $\nu(S) = \mathbb{E}[Y \mid do(S = s)]$ is *non-parametrically* identifiable if, and only if, $P_S(Y)$ is *non-parametrically* identifiable. Trivially, identifiability in the probability terms guarantees identifiability in the expectation (since the expectation's summation or integral is an observational estimand only consisting of identifiable terms $P_s(y)$). Conversely, non-identifiability in the terms $P_S(Y)$ must imply non-identifiability for the expectation, in the absence of further parametrical assumptions (e.g., assuming that $P_s(y) = P_s(-y)$, in which case the terms cancel each other out and make the expectation identifiable[4]). For this reason, we will focus on non-parametric identifiability, and only on queries of the form $P_S(Y)$—which can be determined thanks to the ID algorithm—as a proxy for identifiability of $\nu(S)$.

## G.1. The ID algorithm

Let us provide the necessary background definitions and properties to understand the ID algorithm and follow the proofs of our identifiability criterion.

**Definition G.1.** Let $G = (V \cup U, E)$ be a Directed Acyclic Graph (DAG) with nodes $V \cup U$ (measured and latent nodes, respectively) and edges $E$. Its **latent projection** is an **Acyclic Directed Mixed Graph** (ADMG) $G^* = (V, E^*)$, which is a graph containing directed and bidirected edges and no directed cycles. In particular, $G^*$ only contains the measured nodes in $G$ and: (1) any directed edge in $G$ between measured nodes; (2) an edge between measured nodes if both are connected in $G$

---

[3] A probability distribution is Markov-relative to a DAG $G = (V \cup U, E)$ if $P(V, U) = \prod_{X \in V \cup U} P(X \mid Pa_G(X))$, where $Pa_G(X)$ is the set of parents of $X$ in $G$.

[4] Such parametric cancellations invoke assumptions that hold measure zero over the space of all possible distributions compatible with the graph. Therefore, our proof establishes equivalence only for non-parametric identifiability. We rely strictly on the structural properties of the graph, excluding any further distributional assumptions. This logic mirrors Pearl's argument regarding the equivalence of graph d-separation and conditional independence in almost all distributions ((Pearl, 2009), page 18, following Theorem 1.2.4).

**Algorithm 9** ID$(T, S, G)$

---

**Input:** Disjoint sets $T, S \subseteq \mathcal{V}$, ADMG $G = (\mathcal{V}, E)$.
**Output:** Boolean indicating whether $P_S(T)$ is identifiable.
1: If $S = \emptyset$, **return** TRUE.
2: If $\mathcal{V} \neq An_G(T)$, **return** ID$(T, S \cap An_G(T), G[An_G(T)])$.
3: Let $W := (\mathcal{V} \setminus S) \setminus An_{G_{\overline{S}}}(T)$. If $W \neq \emptyset$, **return** ID$(T, S \cup W, G)$.
4: If $\mathcal{V} \setminus S \notin \mathcal{C}(G[\mathcal{V} \setminus S])$, **return** $\wedge_{C_i \in \mathcal{C}(G[\mathcal{V} \setminus S])}$ ID$(C_i, \mathcal{V} \setminus C_i, G)$.
5: If $\mathcal{V} \in \mathcal{C}(G)$, **return** FALSE.
6: If $\mathcal{V} \setminus S \in \mathcal{C}(G)$, **return** TRUE.
7: Let $C \in \mathcal{C}(G)$ s.t. $C \supset \mathcal{V} \setminus S$, **return** ID$(T, S \cap C, G[C])$.

---

by a directed path where all intermediate nodes are in $U$; and (3) a bidirected edge $V_i \leftrightarrow V_j$ between measured nodes if both are connected by a path of the form $V_i \leftarrow \cdots \leftarrow \cdot \rightarrow \cdots \rightarrow V_j$ where all intermediate nodes are in $U$.

In the following, when talking about a graph, we refer to the ADMG latent projection of the DAG under study.

**Notation.** Let $G$ be an ADMG. We denote by $V(G)$ the set of nodes of $G$. We denote by $G[S]$ the induced subgraph of $G$ filtering to the nodes $S \subseteq V(G)$ and only preserving those edges connecting preserved nodes. For a node $Y \in V(G)$, we denote by $An_G(Y)$ the set of (directed) ancestors of node $Y$ in $G$ including $Y$, and by $De_G(Y)$ the set of (directed) descendants of node $Y$ in $G$ including $Y$. We overload the notation with the ancestors of a set of nodes $T \subseteq V(G)$, $An_G(T) := \bigcup_{t \in T} An_G(t)$, and equivalently for $De_G(T)$. We denote the intervened graph $G_{\overline{S}}$ as the subgraph of $G$ such that any incoming edge to the nodes in $S$ is removed. For this reason, in this section, we will avoid denoting the closure of a coalition $S$ by $\overline{S}$ to avoid ambiguity.

**Definition G.2.** The **root set** $R$ of an ADMG $G$ is the set of nodes in $G$ with no proper descendants: $R := \{X \in V(G) \mid De_G(X) = \{X\}\}$.

**Definition G.3.** Let $G$ be an ADMG. We say that a set of nodes $V$ is a **C-component** within an ADMG $G$ if all nodes are connected via bidirected arcs. We denote the set of maximal C-components of a graph $G$ by $\mathcal{C}(G)$. In particular, $\mathcal{C}(G)$ constitutes a partition of $V(G)$. We also say $G$ is a C-component when $V(G)$ is a C-component in $G$.

**Definition G.4.** We say an ADMG $G$ is an $R$-rooted **C-forest** if $R$ is its root set, $G$ is a C-component, and all its nodes have at most one (directed) child.

**Definition G.5.** Let $G$ be an ADMG, and $S, T$ disjoint subsets of variables in $V(G)$. We say a pair $(F, F')$ is an $R$-rooted **hedge** for $P_S(T)$ if both $F$ and $F'$ are $R$-rooted C-forests such that $F' \subseteq F \subseteq G$, $F \cap S \neq \emptyset$, $F' \cap S = \emptyset$ and $R \subseteq An_{G_{\overline{S}}}(T)$.

Hedges are the reason for non-identifiability of causal effects. See the following theorem from (Shpitser & Pearl, 2006) (Theorem 4).

**Theorem G.6.** *If $(F, F')$ is a hedge for $P_S(T)$ in an ADMG $G$, $P_S(T)$ is not identifiable in $G$.*

We are now ready to study the ID algorithm (Shpitser & Pearl, 2006), with which we can ascertain the identifiability of any query $P_S(T)$. We present a simplified version in Algorithm 9.

### G.2. Identifiability criterion

When evaluating do-SHAP, one must evaluate causal queries $\nu(S) := \mathbb{E}[Y \mid do(S = s)]$ for multiple coalitions $S \subseteq [d]$. If one such query is not identifiable, we cannot continue evaluating do-SHAP and must raise an error. However, this forces practitioners to execute do-SHAP without knowing if an estimate can be raised, while also running the ID algorithm for every single coalition that must be evaluated.

Instead, we present the following result, which allows us to determine the identifiability of all $2^d$ coalition values $\nu(S)$ just by determining the identifiability of the $d$ singleton coalitions $\{i\} \subseteq [d]$.

**Theorem G.7.** *Let $G$ be an ADMG over $\mathcal{V}$, with a single $Y \in \mathcal{V}$ target node. If $\exists X \subseteq \mathcal{V} \setminus \{Y\}$ such that $P_X(Y)$ is not identifiable, then $\exists s \in X \cup (\mathcal{V} \setminus An_{G_{\overline{X}}}(Y))$ such that $P_{\{s\}}(Y)$ is not identifiable.*

We will devote the remainder of this section to proving this theorem. Let us first prove a set of lemmas that will progressively build towards the proof.

**Lemma G.8.** *Let $G$ be an ADMG over $\mathcal{V}$, with a set $Y \subseteq \mathcal{V}$ the target nodes. If $\exists X \subseteq \mathcal{V} \setminus Y$ such that $P_X(Y)$ is not identifiable, then the* ID *algorithm ends at line 5 in a recursive call* $\text{ID}(T, S, G')$ *for $S$ and $T$ non-empty disjoint subsets of $\mathcal{V}' \subseteq \mathcal{V}$, and $G'$ an induced subgraph $G' = G[\mathcal{V}'] \subseteq G$. Then, there exists $R$-rooted C-forests $(F, F')$, $R \subseteq \mathcal{V}'$ such that they constitute a hedge for $P_S(T)$.*

*Proof.* Firstly, the ID algorithm (Shpitser & Pearl, 2006) always terminates (Lemma 3 in their paper), it is sound (Theorem 5) and complete (Corollary 2). Since we assume that $P_X(Y)$ is not identifiable, it must be that when we evaluate $\text{ID}(Y, X, G)$, we will eventually return FALSE from line 5 at a certain recursion level $\text{ID}(T, S, G')$. Both $X$ and $S$ must be non-empty (otherwise line 1 would have returned TRUE). Both $S$ and $T$ must be subsets of $\mathcal{V}$ and disjoint (since all recursive calls maintain them being disjoint given the initial assumption that $Y \cap X = \emptyset$). Finally, $G'$ is an induced subgraph $G' = G[\mathcal{V}'] \subseteq G$, since all recursive calls will at most remove nodes from $G$.

Let $R$ be the root set of $G'[\mathcal{V}' \setminus S]$. In particular, $R \cap S = \emptyset$. Let $F'$ be an edge subgraph of $G'[\mathcal{V}' \setminus S]$ such that its root set remains $R$ but all observable nodes have at most one child, and all confounding arcs in $G'[\mathcal{V}' \setminus S]$ are present. Note that $G'[\mathcal{V}' \setminus S]$ is a C-component (otherwise, line 4 would have triggered) and in creating $F'$ we did not remove any confounding arcs, so $F'$ is also a C-component, and therefore an $R$-rooted C-forest. Now, let us define an edge subgraph $F \subseteq G'$ by starting from $F'$, adding all nodes in $S$ and, for every such node adding only one edge to one of its children in $G'$, as well as all bidirected edges. Since every new node has a child in $V(F)$, the root set remains $R$, and $F$ is an $R$-rooted C-forest by the same reasoning as before because line 5 guarantees $\mathcal{C}(G') = \{\mathcal{V}'\}$. Additionally, $F' \subseteq F$, $V(F') \cap S = \emptyset$ and $V(F) \cap S = S \neq \emptyset$. All that remains is to prove that $R \subseteq An_{G'_{\overline{S}}}(T)$. First, note that $R \subseteq An_{G'}(T)$ by line 2, so there exist paths from nodes $r \in R$ to nodes $t \in T$. Let us assume that one such $r$ has all its directed paths blocked by $S$. Then, $r \notin An_{G_{\overline{S}}}(T)$ and $r \notin S$ (because $R \cap S = \emptyset$), which means that $W$ in line 3 would have contained $r$. By contradiction, $R \subseteq An_{G'_{\overline{S}}}(T)$ and, finally, $(F, F')$ is a hedge for $P_S(T)$ in $G'$. $\qquad \square$

We now define some notation that will guide the proofs that follow. Let $G$ be an ADMG over $\mathcal{V}$, with a *single* target node $Y \in \mathcal{V}$. If $\exists X \subseteq \mathcal{V} \setminus \{Y\}$ such that $P_X(Y)$ is not identifiable, let us focus on a single call-path returning FALSE, and denote the $i$-deep call $ID(T^i, S^i, G^i)$ (with corresponding $\mathcal{V}^i := V(G^i)$) along the call-path, starting on $ID(\{Y\}, X, G)$ (with $\{Y\} = T^0, X = S^0, G = G^0$) and ending at depth $d \geq 0$ in line 5 for the found hedge. Whenever line 4 or 7 are called at depth $i$, let us denote by $C^i$ the C-component that appears either: in line 4, for the recursive call to ID in the particular branch following the call-path, or the one in line 7 that filters $S$. Let $d'$ be the first depth level at which line 4 is triggered or $d + 1$ if it never does.

**Lemma G.9.** *If line 7 triggers at $ID(T^i, S^i, G^i)$, then $\mathcal{V}^i \setminus S^i \subsetneq C^i \subsetneq \mathcal{V}^i$. In particular, $S^{i+1} \subsetneq S^i$.*

*Proof.* $\mathcal{V}^i \setminus S^i \subset C^i \subset \mathcal{V}^i$ and it is a maximal C-component in $G^i$ by construction. If $C^i = \mathcal{V}^i \setminus S^i$, then $\mathcal{V}^i \setminus S^i \in \mathcal{C}(G^i)$ and line 6 would have triggered. If $C^i = \mathcal{V}^i$, then $V^i \in \mathcal{C}(G)$ and line 5 would have triggered. Finally, since $C^i \neq \mathcal{V}^i$ and $C^i \supsetneq \mathcal{V}^i \setminus S^i$, then $S^i \cap C \subsetneq S^i$. $\qquad \square$

**Lemma G.10.** *Some observations that will become relevant throughout the following discussion are:*

- *Line 1 never triggers along the failing call-path, and $\forall i \leq d, S^i \neq \emptyset$; otherwise, the call-path would not end at line 5.*

- *Line 6 never triggers along the failing call-path; otherwise, the call-path would not end at line 5.*

- *Line 5 does not trigger along the failing call-path for depths $i < d$.*

- *$G$ can only have nodes removed: $G^{i+1} \subseteq G^i$, $\forall i < d$.*

- *We can only remove nodes from the local set of intervened variables $S$ on lines 2 and 7, while also removing them from the local graph $G$.*

- *We can only add nodes to the local set of intervened variables $S$ on lines 3 and 4, while also removing them from the set of unused nodes $\mathcal{U} := \mathcal{V} \setminus (T \cup S)$.*

- *The set of unused nodes can never grow: $\mathcal{U}^{i+1} \subseteq \mathcal{U}^i$, $\forall 0 \leq i < d$.*

- *T does not change until line 4 is called:* $\forall i \leq d', T^i = \{Y\}$.

**Lemma G.11.** *Under the previous assumptions, line 4 can trigger at most once.*

*Proof.* Assuming that line 4 triggers at least once, let $d' < d$ be the depth of its first call, and $C^{d'}$ the C-component whose recursive call we follow along the call-path. Right after that, $T^{d'+1} = C^{d'}$, $S^{d'+1} = \mathcal{V}^{d'} \setminus C^{d'}$, and $U^{d'+1} = \mathcal{V}^{d'+1} \setminus (T^{d'+1} \cup S^{d'+1}) = \emptyset$. Assume there is a minimal $d'' > d'$ in which line 4 triggers again. Then, since only line 4 can change $T$, $T^{d''} = T^{d'+1} = C^{d'}$. Note that $\mathcal{V}^{d''} \setminus S^{d''} = T^{d''} = C^{d'}$ (otherwise $U^{d''}$ would increase). Therefore, $\mathcal{C}(G^{d''}[\mathcal{V}^{d''} \setminus S^{d''}]) = \mathcal{C}(G^{d''}[C^{d'}]) = \mathcal{C}(G^{d'}[C^{d'}]) = \{C^{d'}\}$ since $G^{d''} \subseteq G^{d'}$ and $C^{d'}$ is a maximal C-component in $G^{d'}[\mathcal{V}^{d'} \setminus S^{d'}] \supsetneq G^{d'}[C^{d'}]$, which means that line 4 cannot trigger again. In particular, $T^i = C^{d'}$, $\forall i : d' < i \leq d$. $\qquad \square$

**Lemma G.12.** *Under the previous assumptions, if $\mathcal{V} = An_G(Y)$ and $X$ is closed (i.e., $X = X \cup (\mathcal{V} \setminus An_{G_{\overline{X}}}(Y))$), then line 3 never triggers.*

*Proof.* By the fact that the closure of a closed set is itself, line 3 cannot trigger at depth 0. Let us prove the statement by induction.

Firstly, consider a depth $i$, $0 < i < d'$, in which line 3 is triggered, while assuming it has not been triggered before. From Lemma G.10, $T^i = \{Y\}$. For depth $i$, it must be that $W^i := (\mathcal{V}^i \setminus S^i) \setminus An_{G^i_{\overline{S^i}}}(Y) \neq \emptyset$, with nodes $w \notin S^i$ and $w \notin An_{G^i_{\overline{S^i}}}(Y)$, but $w \in An_{G^i}(Y)$ (since we skip line 2 to trigger 3), so there are directed paths from $w$ to $Y$, all blocked by $S^i$. Let us now study depth $i - 1$. Since $w \notin S^i$ and $S^{i-1} \supseteq S^i$, if $w \in S^{i-1}$ it must have been removed by lines 2 or 7, but that would also remove it from $\mathcal{V}^i$, which is not the case, so we know that $w \notin S^{i-1}$. Additionally, if we assumed that $w \notin An_{G^{i-1}_{\overline{S^{i-1}}}}(Y)$, since $w \notin S^i$, then $w \in W^{i-1}$ but because line 3 did not trigger at depth $i - 1$, $W^{i-1} = \emptyset$. Consequently, $w \in An_{G^{i-1}_{\overline{S^{i-1}}}}(Y)$ and $w \notin S^{i-1}$, so there are directed paths from $w$ to $Y$ unblocked by $S^{i-1}$, going through nodes in a set $Z \subseteq \mathcal{V}^{i-1} \setminus S^{i-1}$, which all must have disappeared from $G^i$ to make it so $w \notin An_{G^i_{\overline{S^i}}}(Y)$. Since line 2 preserves all directed paths to $Y$, this can only happen if line 7 was triggered at depth $i - 1$. However, this cannot be the case either: the C-component $C^{i-1} \supsetneq \mathcal{V}^{i-1} \setminus S^{i-1}$ preserves all non-intervened nodes, which is the case for $Z$. Therefore, it must be that our initial assumption was false, $W^i$ is empty, and so line 3 was not triggered at depth $i < d'$ after all.

For depth $d'$, we naturally skip line 3 to reach line 4. Afterwards, $\mathcal{V}^{d'+1} = T^{d'+1} \sqcup S^{d'+1}$ so $\forall d' < i \leq d, \mathcal{U}^i := \mathcal{V}^i \setminus (T^i \cup S^i) = \emptyset$ (Lemma G.10), so $W^i = \emptyset$, and line 3 is never called either. $\qquad \square$

**Lemma G.13.** *Let $G$ be an ADMG over $\mathcal{V}$, with a single $Y \in \mathcal{V}$ target node. If $\exists X \subseteq \mathcal{V} \setminus \{Y\}$ such that $P_X(Y)$ is not identifiable, let $\mathtt{ID}(T^d, S^d, G^d)$ be the last call resulting in the hedge given by Lemma G.8. Then $S^d \cap (X \cup (\mathcal{V} \setminus An_{G_{\overline{X}}}(Y))) \neq \emptyset$.*

*Proof.* Let $X'$ be the closure of $X$, $X' := X \cup (\mathcal{V} \setminus An_{G_{\overline{X}}}(Y))$ (here denoted $X'$ instead of $\overline{X}$ to avoid ambiguity of notation), and let us prove that $S^d \cap X' \neq \emptyset$. Note that if line 2 or 3 trigger on the first calls, we end up in a recursive call $\mathtt{ID}(\{Y\}, S^i, G^i)$ where $\mathcal{V}^i = An_G(Y)$ and $S^i = An_G(Y) \cap (X \cup (\mathcal{V} \setminus An_{G_{\overline{X}}}(Y)))$, so we can assume without loss of generality that $\mathcal{V} = An_G(Y)$ and $X' = X$ at the start of the call-path. Consequently, $S^0 \cap X' = X' \neq \emptyset$ trivially. Let us proceed by induction, proving that $\forall i < d, S^i \cap X' \neq \emptyset$ implies that $S^{i+1} \cap X' \neq \emptyset$, and then so will be the case for $S^d$, which will prove the lemma.

Let us now consider any depth level $0 < i \leq min(d, d')$ and assume that $S^{i-1} \subseteq X'$. Let us study what happens at depth $i - 1$. Lines 1, 3, 4, and 6 cannot trigger (by Lemmas G.10 and G.12 and the fact that $i \leq d'$). If line 2 were to remove any nodes, it would still result in a non-empty proper subset $S^i \subsetneq S^{i-1} \subseteq X'$. If line 7 triggered, $S^i := S^{i-1} \cap C^{i-1} \subsetneq S^{i-1} \subseteq X'$ and $S^i \neq \emptyset$ by Lemma G.9. In all cases, the next recursive $S$ is not empty and only contains nodes in $X'$: $\emptyset \neq S^{i+1} \subsetneq S^i \subseteq X'$.

If line 4 never triggers, we have already proved the result. Otherwise, $d' < d$, and $C^{d'}$ is the C-component whose recursive call we follow along the call-path. In particular, $\mathcal{V}^{d'+1} = \mathcal{V}^{d'}, T^{d'+1} = C^{d'}, S^{d'+1} = \mathcal{V}^{d'} \setminus C^{d'} \supseteq S^{d'}$, so it contains elements of $X'$ and outside $X'$ (because the nodes in $X' \setminus S^{d'}$ have been removed from $G^{d'}$ as well). Additionally, $\forall i > d', \mathcal{U}^i = \emptyset$, and can never increase.

For $d' \leq i \leq d$, let us define $S^i = A^i \sqcup B^i \neq \emptyset$, with $A^i := S^i \cap X'$ and $B^i := S^i \setminus X'$. Note that $A^{d'} = S^{d'}, B^{d'} = \emptyset$, and $A^{d'+1} = S^{d'}, B^{d'+1} = (\mathcal{V}^{d'} \setminus C^{d'}) \setminus A^{d'} \neq \emptyset$, since $S^{d'+1} = \mathcal{V}^{d'} \setminus C^{d'} \supsetneq S^{d'} \subseteq X'$. Additionally, $\forall i > d'+1$, $A^i \subseteq A^{i-1}$ and $B^i \subseteq B^{i-1}$ since we can only remove nodes at this point, given that line 3 and line 4 will not run again (Lemmas G.12 and G.11). Let us prove that if $\forall i : d' < i < d$, $A^i \neq \emptyset$, then $A^{i+1} \neq \emptyset$, for which we must check lines 2 and 7.

If line 2 was triggered at step $i$ and $A^{i+1} = \emptyset$, the next recursion level would have $T^{i+1} = T^i$, $S^{i+1} = B^{i+1} \neq \emptyset$ and $G^{i+1} = G^i[An_{G^i}(T^i)] \subseteq G^i$. At level $i + 1$, line 6 would trigger: neither line 3 nor line 4 can trigger again; if we assume that line 5 triggers, then $\mathcal{V}^{i+1} \in \mathcal{C}(G^{i+1})$, but $B^{i+1} \subseteq B^{d'+1}$ and no node in $B^{d'+1}$ belonged in the same maximal C-component as $C^{d'}$ in $G^{d'}[\mathcal{V}^{d'} \setminus A^{d'}]$, then all bidirected paths from $B^{i+1}$ to $C^{d'}$ were blocked by $A^{d'}$, and now that $A^{i+1} = \emptyset$, these paths are cut; finally, for line 6, if $\mathcal{V}^{i+1} \setminus S^{i+1} = T^{i+1} = C^{d'}$ were not a maximal C-component of the induced subgraph $G^{i+1} = G^i[An_{G^i}(T^i)]$, it must be that a node $b \in S^{i+1} = B^{i+1}$ is connected through a bidirected path to $C^{d'}$ in this graph, which we have just proved to be false, hence proving that the call would return TRUE at line 6. By contradiction, this proves that line 2 would have not removed all of $A^{d'+1} \subseteq X'$.

Finally, if line 7 was triggered at step $i$ and $A^{i+1} = \emptyset$, when before it was not, necessarily $B^{i+1} \neq \emptyset$, and $\forall c \in T^i = C^{d'}$, $\forall b \in B^{i+1} = B^i \cap C^i$, there are bidirected paths connecting $c$ and $b$ in $G^i \supseteq G^{i+1}$, and all are intersected by $S^{d'} = A^{d'}$ (otherwise $b \in C^{d'}$). Consequently, either these intersecting nodes are in $A^i$, in which case they would not have been removed from $A^{i+1}$, or they are not in $A^i$, in which case $b$ would have been removed as well for its paths would be cut. Both cases contradict the assumption that $A^{i+1} = \emptyset$.

Bringing everything together and by induction, $\emptyset \subsetneq A^d \subseteq X'$, proving the lemma. $\qquad \square$

We now have all we need to prove the theorem.

*Proof of Theorem G.7.* Finally, let us finish the proof by showing that if $(F, F')$ is the $R$-rooted hedge for $P_S(T)$ found at the end of the call-stack, $\forall s \in S \cap X'$, with $X' := X \cup (\mathcal{V} \setminus An_{G_{\overline{X}}}(Y))$ the closure of $X$, then $(F, F')$ is also a hedge for $P_{\{s\}}(Y)$. Note that the hedge still fulfills the conditions $F' \subseteq F, F \cap \{s\} \neq \emptyset, F' \cap \{s\} = \emptyset$, both $R$-rooted, both C-forests. So now we only need to check that $R \subseteq An_{G_{\overline{s}}}(Y)$, but if one such $r \in R$ were not in $An_{G_{\overline{s}}}(Y) \supseteq An_{G_{\overline{X'}}}(Y)$, then $r$ would belong in $X'$ by definition of the closure of $X$ and the fact that the closure of a closure is itself. Additionally, throughout the call-stack, $r$ would have never left the set of intervened nodes without leaving the graph itself, hence $r \in S$, contradicting the fact that $S \cap R = \emptyset$ by construction (see Lemma G.8). Therefore, it must be that $R \subseteq An_{G_{\overline{s}}}(Y)$, proving that $(F, F')$ is a hedge for $P_{\{s\}}(Y)$. By Theorem 4 in (Shpitser & Pearl, 2006), $P_{\{s\}}(Y)$ is not identifiable. $\qquad \square$

**Corollary G.14.** *Any do-Shapley value $\phi_i$ for $i \in [d]$ is (non-parametrically) identifiable if, and only if, all singleton coalition probabilities $(P_{\{k\}}(Y))_{k \in [d]}$ are (non-parametrically) identifiable.*

*Proof.* If all singleton coalitions $\{k\} \subseteq [d]$ have identifiable probability terms $P_{\{k\}}(Y)$, there cannot be any coalition $S \subseteq [d]$ for which $P_S(Y)$ is not identifiable (otherwise, by Theorem G.7, $\forall k \in S \cup (\mathcal{V} \setminus An_{G_{\overline{S}}}(Y))$, $P_{\{k\}}(Y)$ would not be identifiable). Then, all the respective $\nu(S) = \mathbb{E}[Y \mid do(S = s)]$ are identifiable, and consequently, $\phi_i = \sum_{j=1}^{r} \nu(c_j) w_i(c_j)$ is also identifiable.

Conversely, and following the non-parametric argument from Footnote 4, if one such $P_{\{k\}}(Y)$ were not identifiable, its $\nu(\{k\})$ term cannot be either. By Equation (4), $\phi_i$ is a weighted sum over all equivalence classes. For the singleton $\{k\}$, let $c_k$ be the class containing the singleton, whose value equals $\nu(\{k\})$ (since $\nu$ is constant within its class). Note that the associated weight $w_i(c_k)$ in Equation (4) is non-zero since $\{k\}$ is a basis of $c_k$. Therefore, the value $\nu(c_k)$, which is non-identifiable, intervenes in the formula for $\phi_i$ with non-zero weight, making $\phi_i$ non-parametrically not identifiable. $\qquad \square$

# H. Additional Experiments

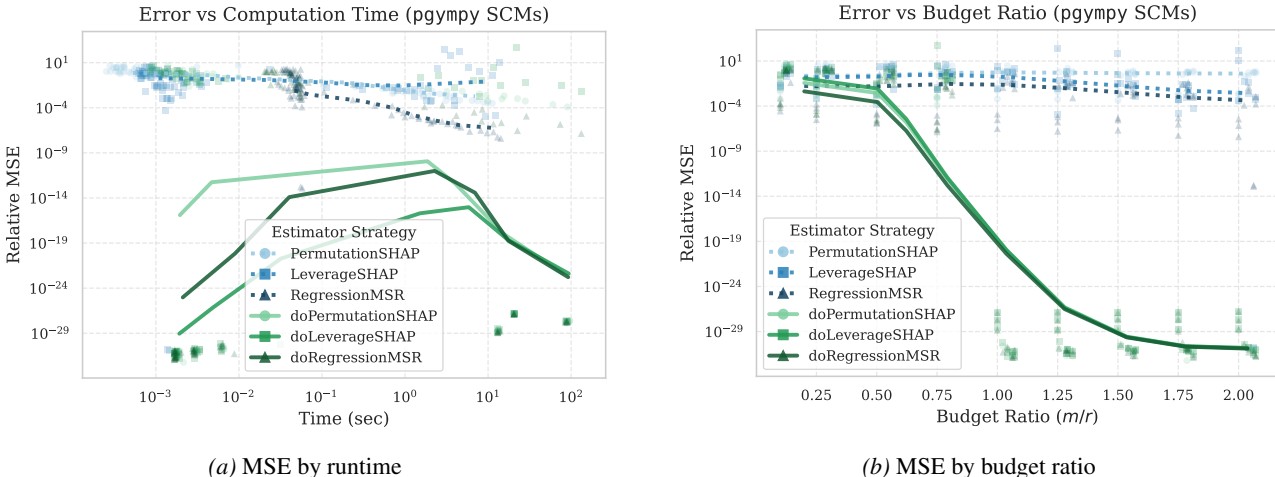

*(a)* MSE by runtime

*(b)* MSE by budget ratio

*Figure 8.* **Estimator Convergence on Posited SCMs.** (a) Comparison of relative MSE by runtime, with median lines on SCMs from the `pgmpy` package (Ankan & Textor, 2024). The doEstimator variants give much lower error even on a time-adjusted comparison with SOTA Shapley value estimators employing the FRA caching of Parafita et al. (2025). (b) Comparison of relative MSE by budget ratio $m/r$. The doEstimators achieve machine precision when $m \geq r$, and start substantially outperforming the SOTA estimators as $m$ approaches $r$.

*Table 2.* The number of new classes discovered per query by each algorithm on the learned SCMs. As intended by their design, the doEstimators discover exactly one new class per query via the BoundarySampler.

| Estimator | Mean | Std. Dev. |
|---|---|---|
| PermutationSHAP | 0.517 | 0.254 |
| LeverageSHAP | 0.73 | 0.259 |
| RegressionMSR | 0.73 | 0.259 |
| doPermutationSHAP | 1 | 0 |
| doLeverageSHAP | 1 | 0 |
| doRegressionMSR | 1 | 0 |

*Table 3.* The number of new classes discovered per second by each algorithm on the learned SCMs. While comparable, the SOTA Shapley value estimators are faster because they avoid the graph exploration, and hence generally discover more new classes per second. However, because the doEstimators exploit the structure of the do-Shapley value, they achieve much more accurate estimates, as shown in Figure 5a and 8a.

| Estimator | Mean | Std. Dev. |
|---|---|---|
| PermutationSHAP | 327 | 200 |
| LeverageSHAP | 324 | 201 |
| RegressionMSR | 193 | 65.9 |
| doPermutationSHAP | 241 | 162 |
| doLeverageSHAP | 303 | 180 |
| doRegressionMSR | 240 | 155 |

*Table 4.* The number of new classes discovered per query by each algorithm on the `pgmpy` SCMs. The results are consistent to the learned SCMs: As intended by their design, the doEstimators discover exactly one new class per query via the BoundarySampler.

| Estimator | Mean | Std. Dev. |
|---|---|---|
| PermutationSHAP | 0.534 | 0.42 |
| LeverageSHAP | 0.789 | 0.503 |
| RegressionMSR | 0.789 | 0.503 |
| doPermutationSHAP | 1 | 0 |
| doLeverageSHAP | 1 | 0 |
| doRegressionMSR | 1 | 0 |

*Table 5.* The number of new classes discovered per second by each algorithm on the `pgmpy` SCMs. Even though evaluating $\nu$ is faster, the results are consistent to the learned SCMs: While comparable, the SOTA Shapley value estimators are faster because they avoid the graph exploration, and hence generally discover more new classes per second. However, because the doEstimators exploit the structure of the do-Shapley value, they achieve much more accurate estimates, as shown in Figure 5a and 8a.

| Estimator | Mean | Std. Dev. |
|---|---|---|
| PermutationSHAP | 7.65e+03 | 3.89e+03 |
| LeverageSHAP | 7.64e+03 | 3.5e+03 |
| RegressionMSR | 1.04e+03 | 1.41e+03 |
| doPermutationSHAP | 4.31e+03 | 3.05e+03 |
| doLeverageSHAP | 4.65e+03 | 3.17e+03 |
| doRegressionMSR | 3.6e+03 | 3.85e+03 |

# I. Extensions of the Shapley value

The Shapley value was extended in multiple ways. Semivalues (Dubey et al., 1981) , such as the Banzhaf value (Banzhaf III, 1964), extend the Shapley values to alternative weighting schemes $(q_s)_{s=0,\ldots,n-1} \geq 0$ as

$$\phi_i^q = \sum_{S \subseteq [d] \setminus \{i\}} [\nu(S \cup \{i\}) - \nu(S)] q_{|S|}. \tag{12}$$

If $q$ is a probability distribution over $2^{[d]}$, then they are also referred to as cardinal-probabilistic values (Fujimoto et al., 2006). By replacing the weights in Equation (4), we can derive an efficient computation for any semivalue.

Another line of work extends the Shapley value to higher-order interactions, known as the Shapley interaction index (Grabisch & Roubens, 1999), and defined by

$$\phi_U = \sum_{S \subseteq [d] \setminus U} \Delta_U(S) p_{|S|}^{|U|} \text{ with } \Delta_U(S) := \sum_{L \subseteq U} (-1)^{|U|-|L|} \nu(S \cup L),$$

and $p_s^u := \frac{1}{(d-u+1) \cdot \binom{d-u}{s}}$. The discrete derivative $\Delta_U(S)$ thereby measures the interaction of $U$ in the presence of $S$, and directly extends the marginal contribution. We now utilize a result by Zern et al. (2023b)[Proposition 1] to efficiently compute the Shapley interaction index for value functions of the shape $\mathbf{1}[\underline{S} \subseteq S \subseteq \overline{S}]$.

**Proposition I.1** (Zern et al. (2023a)). *Given subsets $\underline{S} \subseteq \overline{S} \subseteq [d]$ and value function $\nu(S) = \mathbf{1}[\underline{S} \subseteq S \subseteq \overline{S}]$, the Shapley interaction index is given by the weights $\omega_{a,b} := \frac{1}{a+b+1} \binom{a+b}{a}^{-1}$ and*

$$\phi_U = (-1)^{|U \cap ([d] \setminus \overline{S})|} \omega_{|\underline{S}| - |\overline{S} \cap U|, |[d] \setminus (\overline{S} \cup U)|},$$

*if $U \subseteq A \cup ([d] \setminus B)$, and $\phi_U = 0$ otherwise.*

Notably, the weights $\omega$ can be precomputed. Consequently, we obtain the following result by the linearity of the Shapley interaction index (Grabisch & Roubens, 1999).

**Proposition I.2.** *The do-Shapley interaction index is given by*

$$\phi_U = \sum_{j=1}^{r} \nu(c_j) (-1)^{|U \cap ([d] \setminus \overline{S}_j)|} \omega_{|\underline{S}_j| - |\overline{S}_j \cap U|, |[d] \setminus (\overline{S}_j \cup U)|}.$$

*Proof.* We define $\nu_j(S) := \mathbf{1}[\underline{S}_j \subseteq S \subseteq \overline{S}_j]$ for each irreducible set from $j = 1, \ldots, r$. Furthermore, we denote $\phi_U[\nu]$ as the Shapley interaction index with respect to $\nu$. Then, by linearity (Grabisch & Roubens, 1999) of $\phi_U$, we obtain

$$\phi_U[\nu] = \phi_U\left[\sum_{j=1}^{r} \nu(c_j) \cdot \mathbf{1}[\underline{S}_j \subseteq S \subseteq \overline{S}_j]\right] = \sum_{j=1}^{r} \nu(c_j) \phi_U[\nu_j] = \sum_{j=1}^{r} \nu(c_j)(-1)^{|U \cap ([d] \setminus \overline{S}_j)|} \omega_{|\underline{S}_j| - |\overline{S}_j \cap U|, |[d] \setminus (\overline{S}_j \cup U)|}.$$

$\square$

With the efficient computation of the Shapley interaction index (Grabisch & Roubens, 1999), we can directly extract the $n$-Shapley values (Lundberg et al., 2018a; Bordt & von Luxburg, 2023) using the following recursion:

$$\Phi_U^n := \begin{cases} \phi_U & \text{if } |U| = n, \\ \Phi_U^{n-1} + B_{d-|U|} \sum_{\substack{K \subseteq [d] \setminus U \\ |K| + |S| = n}} \phi_{U \cup K} & \text{if } |U| < n, \end{cases}$$

with $\Phi_i^1 := \phi_i$ for all $i \in [d]$. The $n$-Shapley values have first been introduced by Lundberg et al. (2018b), and were later generalized to arbitrary order $n$ (Bordt & von Luxburg, 2023). Importantly, they satisfy the generalized efficiency axiom for Shapley interactions (Bordt & von Luxburg, 2023), i.e.

$$\sum_{U \subseteq [d]: |U| \leq n} \Phi_U^n = \nu([d]),$$

where we defined $\Phi_\emptyset^0 := \nu(\emptyset)$.

**do-Shapley Interactions for SCMs.** Generalizations of the Shapley value to higher-order interactions provide a principled framework for explanations with varying degrees of granularity and expressivity. To illustrate this for do-Shapley values and their interaction-based extensions, we compute Shapley interactions using shapiq (Muschalik et al., 2024a) for SCMs derived from four datasets from the TALENT benchmark (Liu et al., 2025). Following Section 6, the SCMs are learned from data using GRaSP (Lam et al., 2022), with the corresponding variables described in Tables 6 to 9. The resulting do-Shapley interactions are visualized in Figures 9 to 12. First-order explanations recover the standard do-Shapley values, while the highest-order interactions correspond to the Möbius transform of the value function. We present the shapiq explanations as overlays on the learned SCMs, where node sizes represent main effects and edge or hyperedge widths encode interaction strength. The color indicates the direction of the effects (blue denotes a negative interaction and red a positive interaction). While first-order do-Shapley values summarize aggregated causal effects of individual features, increasing the interaction order yields progressively finer-grained insights into the causal structure. At the same time, the growing number of interaction terms poses interpretability challenges, motivating application-specific post-processing and selection of relevant interaction orders.

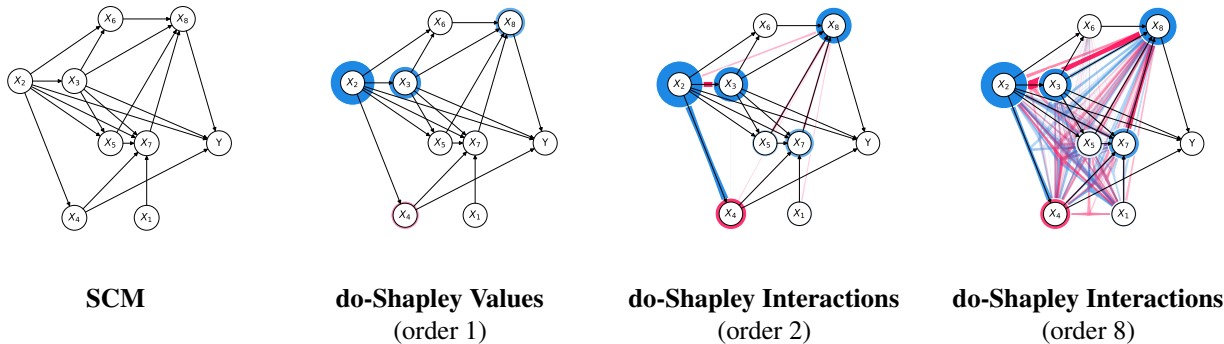

| SCM | do-Shapley Values (order 1) | do-Shapley Interactions (order 2) | do-Shapley Interactions (order 8) |

*Figure 9.* do-Shapley values and interactions for BRAZILIAN_HOUSES_REPRODUCED of increasing order (feature names in Table 6). The size of the nodes and edges (hyperedges) denotes the strength of the effect. The color denotes the direction (blue negative, red positive).

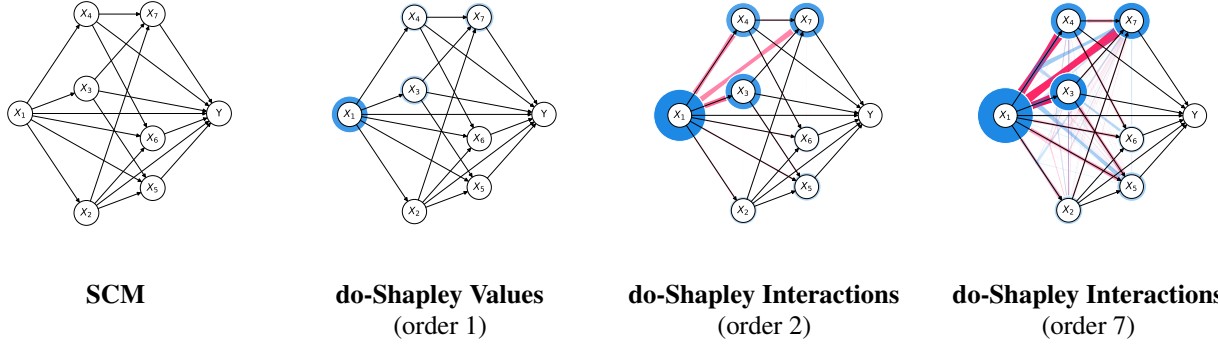

| SCM | do-Shapley Values (order 1) | do-Shapley Interactions (order 2) | do-Shapley Interactions (order 7) |

*Figure 10.* do-Shapley values and interactions for FOREX_AUDJPY-HOUR-HIGH of increasing order (feature names in Table 9). The size of the nodes and edges (hyperedges) denotes the strength of the effect. The color denotes the direction (blue negative, red positive).

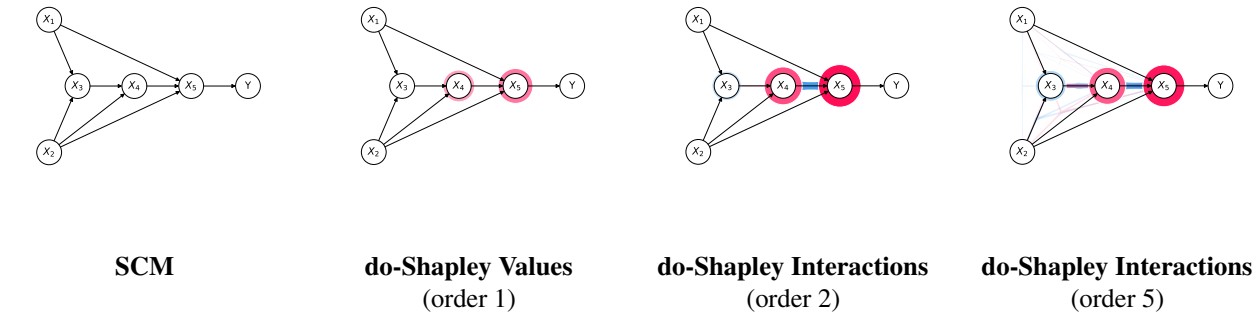

| SCM | do-Shapley Values (order 1) | do-Shapley Interactions (order 2) | do-Shapley Interactions (order 5) |

*Figure 11.* do-Shapley values and interactions for LAPTOP_PRICES_DATASET of increasing order (feature names in Table 8). The size of the nodes and edges (hyperedges) denotes the strength of the effect. The color denotes the direction (blue negative, red positive).

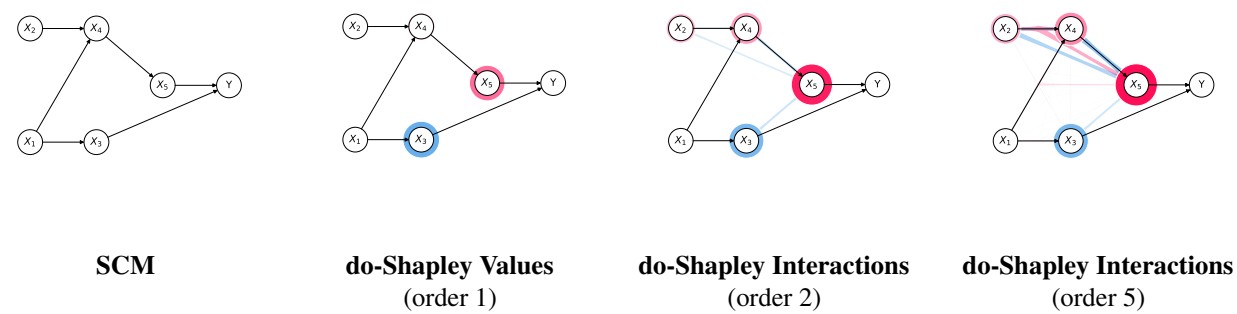

| SCM | do-Shapley Values (order 1) | do-Shapley Interactions (order 2) | do-Shapley Interactions (order 5) |

*Figure 12.* do-Shapley values and interactions for YEAST of increasing order (feature names in Table 7). The size of the nodes and edges (hyperedges) denotes the strength of the effect. The color denotes the direction (blue negative, red positive).

| Node | Feature Name |
| --- | --- |
| $X_1$ | hoa_(BRL) |
| $X_2$ | fire_insurance_(BRL) |
| $X_3$ | parking_spaces |
| $X_4$ | rent_amount_(BRL) |
| $X_5$ | rooms |
| $X_6$ | property_tax_(BRL) |
| $X_7$ | target |
| $X_8$ | bathroom |
| $Y$ | area |

*Table 6.* Variables for BRAZILIAN_HOUSES_REPRODUCED: $d = 8$ input features and 9 nodes in total (including target $Y$).

| Node | Feature Name |
| --- | --- |
| $X_1$ | alm |
| $X_2$ | nuc |
| $X_3$ | vac |
| $X_4$ | gvh |
| $X_5$ | mcg |
| $Y$ | mit |

*Table 7.* Variables for YEAST: $d = 5$ input features and 6 nodes in total (including target $Y$).

| Node | Feature Name |
|------|--------------|
| $X_1$ | N_0 |
| $X_2$ | N_1 |
| $X_3$ | N_5 |
| $X_4$ | target |
| $X_5$ | N_4 |
| $Y$ | N_2 |

*Table 8.* Variables for LAPTOP_PRICES_DATASET: $d = 5$ input features and 6 nodes in total (including target $Y$).

| Node | Feature Name |
|------|--------------|
| $X_1$ | Ask_Low |
| $X_2$ | Bid_Low |
| $X_3$ | Bid_Close |
| $X_4$ | Ask_Open |
| $X_5$ | Ask_Close |
| $X_6$ | Bid_Open |
| $X_7$ | Ask_High |
| $Y$ | Bid_High |

*Table 9.* Variables for FOREX_AUDJPY-HOUR-HIGH: $d = 7$ input features and 8 nodes in total (including target $Y$).

