# OpenReview forum: "Exactly Computing do-Shapley Values"
_ICML.cc/2026/Conference — ICML 2026 regular_

### Official Review · Reviewer_bN3G · 2026-03-09

**Soundness:** 3
**Presentation:** 3
**Significance:** 2
**Originality:** 3
**Overall Recommendation:** 4
**Confidence:** 4

**Summary:**

This paper studies the problem of reducing the computational burden of computing do-Shapley values in Structural Causal Models (SCMs). The main idea is to group subsets into equivalence classes based on their basis and closure, and then rewrite the original Shapley sum over $2^d$ coalitions as a weighted sum over $r$ class-wise quantities. Based on this idea, the authors propose an exact computation procedure that explores the lattice, and they also provide a boundary-based approximation scheme for the case of the limited budget. In addition, the paper claims a theoretical result that connects the identifiability of do-Shapley values to the identifiability of singleton interventions. In the experiments, the authors show that $r$ is smaller than $2^d$ in learned sparse SCMs, and therefore a structure-aware estimator can achieve lower estimation error than existing estimators under a fixed query budget.

**Compliance With Llm Reviewing Policy:**

Affirmed.

**Ethical Review Concerns:**

While reviewing this paper among the papers assigned to me, I found an overlap with another submission I was assigned, in the part discussing related works.

In Section 1.1 of this submission, lines 167 to 190, in the second column, it is written as follows:

> Exactly computing Shapley values is generally feasible only when the underlying model possesses exploitable structure (Rozemberczki et al., 2022). This has led to efficient, model-specific algorithms for decision trees and ensembles, including TreeSHAP (Lundberg et al., 2020; Yu et al., 2022), interventional variants (Zern et al., 2023b), and extensions like TreeSHAP-IQ (Muschalik et al., 2024c). Similar exact methods exist for linear models (Strumbelj & Kononenko, 2014), product-kernel networks (Mohammadi et al., 2025a), Gaussian processes (Mohammadi et al., 2025b), graph neural networks (Muschalik et al., 2025), and KNN-based data valuation (Jia et al., 2019; Wang et al., 2023; 2024). When black-box access precludes exact methods, practitioners rely on model-agnostic estimators (Chen et al., 2023; Muschalik et al., 2024b) such as Monte Carlo sampling (Castro et al., 2009; Kolpaczki et al., 2024b;a; Fumagalli et al., 2023; Wang & Jia, 2023) or regression-based approaches like KernelSHAP (Lundberg & Lee, 2017), which has been enhanced via leverage scores (Musco & Witter, 2025) and interaction support (Fumagalli et al., 2024; Tsai et al., 2023). Finally, exact computation has recently merged with estimation via surrogate modeling (Butler et al., 2025), where auxiliary models are fit to the value function to allow efficient extraction of Shapley values (Witter et al., 2025).

On the other hand, in another submission that I received as a reviewer, there is a paragraph in the discussion of related work that says

> Consequently, exact computation is generally feasible only when the model has exploitable structure (Rozemberczki et al., 2022). This has led to efficient, model-specific algorithms for decision trees ensembles, including TreeSHAP (Lundberg et al., 2020c; Yu et al., 2022) and interaction extensions (Zern et al., 2023; Muschalik et al., 2024c). Similar methods exist for linear (Strumbelj & Kononenko, 2014) and product-kernel models (Mohammadi et al., 2025a), Gaussian processes (Mohammadi et al., 2025b), graph neural networks (Muschalik et al., 2025), and KNN-based data valuation (Jia et al., 2019; Wang et al., 2023; 2024). Without such structure, one must rely on estimation given a query budget m to evaluate f. Because Shapley values are widely deployed, a diverse array of estimators has been developed, ranging from Monte Carlo and permutation sampling methods (Castro et al., 2009; Strumbelj & Kononenko, 2014) to regression-based approaches (Lundberg & Lee, 2017). This work focuses on this general model-agnostic setting, where no specific structure of f is assumed.

As it can be seen, they overlap almost to the level of sentence structure.

**Ethical Review Flag:**

Flag this paper for an ethics review.

**Ethics Expertise Needed:**

["Research Integrity Issues (e.g., plagiarism)"]

**Final Justification:**

After considering the rebuttal, I updated my evaluation positively and raised my score. My main concerns were addressed sufficiently. Overall, I believe the work is enough for acceptance.

**Key Questions For Authors:**

Most of my questions below are based on the weaknesses discussed above.

1. Could you clarify how often the exact computation regime is practically attainable, rather than the $r>m$ approximation regime being the main use case?

2. Could you explain more clearly whether the `SimulatedSampler` is theoretically justified for the full do-Shapley target, and in particular whether it is unbiased or instead should be viewed as a heuristic approximation with possible bias when $r \gg m$?

3. Since `SimulatedSampler` appears to sample only from already discovered classes, could you clarify how undiscovered classes are accounted for and how this differs statistically from a rejection samplings?

4. Given that Figures 12 and 13 show noticeable runtime overhead for `doRegressionMSR` and `doLeverageSHAP`, could you provide an error vs. time comparison?

5. Could you clarify why Figure 5 still shows noticeable error at $m/r=1.00$, even though the text suggests that exact recovery should occur once $m \ge r$?

**Limitations:**

No. The authors do mention some limitations, but in the current version, the practical limitations and the constraints of the approximation regime are not discussed in enough detail.

**Strengths And Weaknesses:**

# Strengths

1. Methodological Formulation

The main strength of this paper is that it approaches the do-Shapley computation problem not by simply proposing a faster estimator, but by revisiting the coalition structure itself. The idea of grouping coalitions using basis and closure, and rewriting the original Shapley sum with $2^d$ terms into $r$ class-wise quantities, is conceptually clean and interesting. In particular, this is not just an implementation trick, but an attempt to theoretically organize the redundancy created by the SCM structure, which gives the paper some originality.

2. Empirical Validation for Justifying Motivation

The experiments also showcase the main computational message of the paper well. In particular, they show how much smaller $r$ is than $2^d$ in learned graphs, and they also show in a relatively consistent way that the structure-aware approach can achieve lower error than the baselines under a fixed query budget. Therefore, the paper gives reasonable empirical support for the direction that ``SCM structure can be used to make do-Shapley computation more efficient”,

3. Practical Identifiability Checking

Finally, it is also a positive point that the paper tries to address identifiability together with computational efficiency. It is important from an application perspective to provide not only a faster computation method, but also a way to quickly check whether do-Shapley computation is feasible in the first place. This could play an important role in simplifying practical feasibility checking.

# Weaknesses

1. Limited Discussion on Approximation

Although this paper presents **exact computation** as the main selling point, in practice it still seems likely that many realistic regimes will need to rely on **approximation**. The main message of Figure 4 is that the original exhaustive complexity $2^d$ can be reduced to $r$. Of course, this reduction itself is meaningful, but from the graph it seems that in many cases the reduction is only about 1/10 of the original size, even under a conservative reading. When we think about the absolute scale, this does not necessarily mean that exact computation has become broadly practical. For example, when $d=20$, we have $2^{20} \approx 10^6$, and even after reducing this to 1/10, it is still around $10^5$. This can still be quite expensive for exact enumeration. Therefore, from a practical point of view, the approximation regime $r>m$ may be much more important than the exact regime $r \le m$. However, the framing of the paper strongly emphasizes the structural goodness of exact computation, while the approximation setting, which may be more important in practice, does not seem to be discussed enough.

Related to this, the discussion and justification for the `SimulatedSampler` in Appendix C are also limited. From the current explanation, this algorithm seems to generate coalitions from already discovered classes and adjust the weights inside them. If so, then it seems that there is no direct sampling for classes that have not yet been discovered. In that case, it is not clear in what sense this kind of restricted-support sampling is justified for the full do-Shapley target. In particular, more explanation is needed on whether this method is unbiased, or whether it is only a heuristic approximation, and if there is bias, how large it is, and what kind of problems may happen when $r \gg m$. From the current description, it is also not clear how fundamentally different this method is, from a statistical point of view, from an approach based on the rejection sampling. As discussed above, approximation is likely to be very important in realistic regimes, and an important key point that affects the practical significance of the paper.

2. Tradeoff between Class Searching and Powerset Sampling

Another important practical concern is that in Figures 12 and 13, the proposed structure-aware estimators, `doRegressionMSR` and `doLeverageSHAP`, consistently have higher computation time than the baselines. From Figure 13, this overhead even looks close to 10 times larger in some cases. The problem is that when we view this together with the result in Figure 4, the practical advantage of the proposed method may be more limited than it first appears. In other words, if the exact complexity reduction is only around 1/10 empirically, while at the same time the approximation method is almost 10 times slower than the baseline, then it is natural to ask whether the overall utility of the proposed method is really compelling. The current paper mainly shows MSE improvement under budget ratio, but in practical comparison, it may be more important to compare which method achieves lower MSE under the same runtime. In particular, since the proposed method has a higher cost per query, fixed-budget comparison alone may not be enough to demonstrate realistic efficiency. At least some comparison from an ``error vs. time” perspective would be needed to judge more convincingly whether the accuracy gain is large enough to justify the sacrifice in runtime.

3. Potential Misunderstanding

Figure 5 is also somewhat confusing in terms of presentation. According to the paper’s description and theory, once $m/r=1$, that is, once $m=r$, all classes should have been found and the method should be able to switch to exact computation. Then one would expect the MSE to drop to around machine precision at that point. However, in Figure 5, there still seems to be noticeable error at $m/r=1.00$, and the curve appears to become close to machine precision only when the ratio moves to around 1.5 or 2.0. So, if we look only at the figure, the exact-recovery threshold looks more like $m>r$ rather than $m \ge r$. If this is due to aggregation, rounding, binning, or implementation details, then this should be explained clearly. Otherwise, readers may become confused about whether the main claim and the empirical figure are actually aligned.

---

> ### Author Rebuttal · Authors · 2026-03-30
>
> We thank the reviewer for highlighting the originality of our insights and the practical benefits of our estimators and identifiability checking. We respond in detail below.
>
> **Rebuttal experiments here: https://shorturl.at/qFxQf**
>
> > *Although this paper presents exact computation as the main selling point, in practice it still seems likely that many realistic regimes will need to rely on approximation. [...] Could you clarify how often the exact computation regime is practically attainable [...]?*
>
> We appreciate this important point and agree that **the approximation regime ($m < r$) is practically significant and deserves emphasis**. Whether exact computation is attainable depends on the budget $m$ and graph sparsity. For path-like graphs (e.g., Figure 3(a) in the paper), $r = O(d)$ and exact computation is readily feasible. For denser graphs in our TALENT experiments, $r$ can be larger. Still, even in the approximation regime ($m < r$), the doEstimator variants substantially outperform the baselines (see Rebuttal Figures 1 and 3). We have revised the paper to better balance the discussion between the exact and approximate regimes.
>
> > *Since SimulatedSampler appears to sample only from already discovered classes, could you clarify how undiscovered classes are accounted for and how this differs statistically from a rejection samplings?*
>
> Great question! Fundamentally, our goal with the doEstimator (Algorithm 3) is to exploit the equivalence class structure of the do-Shapley values even when $r \gg m$. To do this, we proceed in two steps:
>
> 1. In the first step, we sample equivalence classes from the graph using the BoundarySampler and a queue. The BoundarySampler improves over rejection sampling by a) sampling according to the *importance* of each class for the Shapley values (rather than the sheer volume) and b) guaranteeing one query leads to one new class (rather than the many wasted queries rejection sampling would incur as the $m$ approaches $r$).
>
> 2. If $m < r$, we use a SOTA Shapley value estimator. Each of these estimators is either biased or unbiased, and samples subsets in its own way. To facilitate the sampling for these estimators, we “simulate” sampling by restricting to the graph structure discovered by the boundary sampler in Step 1. (Details on how we do this efficiently are in Appendix C.) The key is that these simulated samples $S$ are actually very cheap because we already know $\nu(S)$, which is generally the bottleneck because it requires intervening on and evaluating the structural equations of an SCM. Hence, with a very low time cost, we can feed additional samples to the BaseEstimator.
>
> From a query perspective, these “simulated” samples are free because we are exploiting what we know about the structure of the graph. From a time perspective, they are very cheap. We'd like to note that the SimulatedSampler is *not guaranteed to be unbiased when $m < r$*, because it can only sample from discovered classes. However, because the BoundarySampler discovers classes in order of their weight in the Shapley expression, the most influential classes are found first, and we see empirically that the error diminishes rapidly with $m$.
>
> > *Given that Figures 12 and 13 show noticeable runtime overhead for doRegressionMSR and doLeverageSHAP, could you provide an error vs. time comparison?*
>
> Great question! We provide the error vs time comparison Rebuttal Figure 1. The results suggest that the doEstimator variants of the standard estimators can give orders of magnitude lower error on a per time basis.
>
> **Note:** While preparing rebuttal experiments, we identified and corrected a caching-reset issue in the baseline runtime measurements that had made the non-doEstimators appear faster than they are. The corrected timing results appear in Rebuttal Figure 2 (replacing Figure 13 in the original paper). **After correction, the standard and doEstimators have comparable runtimes**, further strengthening the case for the doEstimator variants.
>
> > *Could you clarify why Figure 5 still shows noticeable error at $m/r=1.00$, even though the text suggests that exact recovery should occur once $m \geq r$?*
>
> Thank you for catching this. You were exactly right: this was caused by an off-by-one error in how we indexed the budget. **We have corrected this, and the updated plots (Rebuttal Figure 2) now show that doEstimators achieve machine precision at $m = r$**, consistent with the theory.
>
> > *While reviewing this paper among the papers assigned to me, I found an overlap with another submission I was assigned, in the part discussing related works.*
>
> We appreciate you flagging this. One of our co-authors is also an author on the other submission. The overlapping related-work paragraph was written by this shared author and reused in part across both papers, just reworded. We understand this is inappropriate regardless of shared authorship, and **the paragraph has now been fully rewritten**. We apologize for this mistake.

---

> > ### Author Rebuttal · Reviewer_bN3G · 2026-04-02
> >
> > Thank you for the thoughtful rebuttal. My main concerns have been fully resolved. In particular, the clarification on the approximation regime and the role of the `SimulatedSampler` was helpful, and the additional rebuttal experiments addressed my questions about the practical tradeoff between error and runtime. I also appreciate the correction of the plotting issue. Accordingly, I have raised my rating.
> >
> > Regarding the potential overlap with another submission, I raised it only because I wanted to make sure that this issue was properly identified and corrected. I appreciate the authors' clarification and their willingness to promptly revise the overlapping parts.

---

### Official Review · Reviewer_Ra1S · 2026-03-10

**Soundness:** 3
**Presentation:** 1
**Significance:** 3
**Originality:** 3
**Overall Recommendation:** 4
**Confidence:** 5

**Summary:**

The goal of the paper is to find an efficient way to compute Shapley values.

**Compliance With Llm Reviewing Policy:**

Affirmed.

**Final Justification:**

Rebuttal has convinced me that the authors are honest and will update the paper with respect to our discussion.
I increased my score.

**Key Questions For Authors:**

- L140, right column: the notion of identifiability is related to causal discoveyr, while causal reasoning is discussed here
- Proof of the use of Rule 3 L220
- Def 2.2 : is it the greatest?
- in the equiavlence class, you should precise the equivalnece relation
- can you provide details about the linear complexity of w_i(c)? To me, it depends on the binomial coefficient, which is linear in d (multiplication of two numbers can also be slower than constant time, but this is less problematic)
- can you details the relation with game theory in Prop 3.2?
- As there is an approximation of the shapley value, I would expect a control of the error (is it asymptotically 0?)
- L1079: you didn't prove the two sense of the equivalence
- Appendix E.2: do you really need to use ID? I am guessing that the use of hedges may be enough, which would simplify Thm 5.1.
- I suspect L361 a quadratic complexity to check for global identifiability. Can you provide details please?  And can you illustrate this complexity in the experimental section?

**Limitations:**

yes

**Strengths And Weaknesses:**

The paper is interesting.
It needs effort to be ready for publication, but if authors answer to my points during the rebuttal I would be happy to accept it.
My first problem is about the writing.
The introduction is very long, and introduce concepts/ideas, which are better defined later. It can be splitted in (several) sections.
Then, most of the proofs are hidden in the text, and it would facilitate the reading if it would have been more formal. For example: does algorithm 1 is sound? What is its complexity? Same questions for Algo 2.
In the experimental section, generated data are in fact based on a real data, and then no ground truth is available. Authors argue that the ground truth is estimated first, but statistically this does not make sense (and will be biased).

---

> ### Author Rebuttal · Authors · 2026-03-30
>
> We thank the reviewer for their detailed feedback and comments. **Due to the space constraints, we respond succinctly below.**
>
> **New experiments: https://shorturl.at/qFxQf**
>
> > *Introduction long…definitions later…*
>
> We appreciate this feedback. We have (1) moved related work and background into dedicated sections, and (2) added formal complexity statements for Algorithms 1 and 2 (see next point).
>
> > *Are algorithms 1/2 sound? Complexity?*
>
> Good questions! Algorithm 2's soundness and complexity are detailed post-Lemma 3.1. For Algorithm 1: it is sound via graph reachability on $G'$ (edges into $S$ removed), finding $\underline{S}$ as nodes with directed paths to $Y$ and $\bar{S}$ as nodes blocked by $S$. Complexity is $O(d+e)$ using standard graph traversal (BFS/DFS) on $G'$ ($d$ nodes, $e$ edges).
>
> >*Generated data in experiments based on real data ... no available ground truth.*
>
> We learned SCMs via GRaSP on TALENT due to the scarcity of ground-truth SCMs. To address your concern, we ran experiments on ground-truth SCMs from ```pgmpy```. **Results are consistent with learned SCMs** (Rebuttal Figures 3 and 4).
>
> > *L140, right column:...identifiability is related to causal discovery…causal reasoning discussed here*
>
> We updated the opening sentences to prevent confusion with causal discovery. Our identifiability discussion assumes a hypothesized graph for causal reasoning; verifying non-parametric identifiability of $\nu(S)$ via ID ensures valid downstream estimations.
>
> > *Proof for Rule 3 L220*
>
> Intuitively, because $T$ is bounded by $\underline{S}$ and $\bar{S}$, intervening on $\underline{S}$ strictly blocks all directed paths from $T \setminus \underline{S}$ to $Y$. This makes any further interventions on $T \setminus \underline{S}$ redundant by Rule 3, yielding $\nu(T) = \nu(\underline{S})$. Symmetric logic applies to $\bar{S}$. **We have added a proof sketch in the appendix**; see also Parafita et al. (2025).
>
> > *Def 2.2: greatest?*
>
> Yes. By Definition 2.2, $\bar{S}$ explicitly collects every node $j \in [d]$ where all directed paths to $Y$ intersect $S$. Any other set meeting this criterion must be a subset of $\bar{S}$, making it the unique, maximal superset.
>
> > *...equivalence class w/ precise the equivalence relation…*
>
> $S \sim T \iff \underline{S} \subseteq T \subseteq \bar{S}$ (equivalently, $\underline{S} = \underline{T}$ and $\bar{S} = \bar{T}$). This is reflexive, symmetric, and transitive, now stated explicitly.
>
> > *Details about the linear complexity of w_i(c)?*
>
> Computing $w_i(c)$ takes $O(d)$ time because Equation 5 sums at most $d$ terms. The binomial coefficients do not require from-scratch computation; they are calculated iteratively in linear time.
>
> > *Details w.r.t. game theory in Prop 3.2?*
>
> Proposition 3.2 connects to cooperative game theory via the *probabilistic value* framework (Weber, 1988). Standard Shapley values correspond to a specific weighting over coalitions; our equivalence-class decomposition preserves these weights exactly; it is a re-indexing of the same sum, not an approximation. Concretely, the weight $w_i(c)$ in our Equation 4 aggregates the standard Shapley coalition weights over all sets in the equivalence class $[c]$, and we derive a closed form for this aggregation. We have added a remark in the revised manuscript to make this connection explicit.
>
> > *As there is Shapley value approximation, I would expect a control of the error…*
>
> When $m \geq r$, error drops to **machine precision** (Rebuttal Figure 2). When $m < r$, guarantees inherit from the BaseEstimator (e.g., PermutationSHAP is unbiased/consistent). **Empirically, even for moderate $m/r$, doEstimator variants achieve orders-of-magnitude lower error** (Rebuttal Figure 1).
>
> > *L1079: didn't prove the two sense of the equivalence*
>
> We agree this deserved clarification. Our proof establishes both directions under standard *faithfulness* (excluding measure-zero parametric cancellations), mirroring Pearl (2009). The forward direction follows from the ID algorithm, the reverse from the hedge construction.  **We have updated L1079 to state this assumption explicitly.**
>
> > *Appendix E.2: do you really need to use ID? …hedges may be enough….*
>
> We appreciate this suggestion. While it could simplify the proof, the Hedge Criterion has required corrections (Shpitser, 2023), we therefore prove Theorem 5.1 directly to ensure correctness. We agree, however, that exploring a Hedge-based simplification is an interesting direction.
>
> > *I suspect a quadratic complexity for global identifiability…*
>
> The ID algorithm's exact runtime is polynomial, depending heavily on graph density. Our core contribution reduces the ID calls from exponentially many to $d$ (just the singletons); running ID $d$ times is computationally negligible compared to evaluating $\nu$. In our experiments, the SCMs learned via GRaSP are Markovian (no unobserved confounders), which trivially guarantees identifiability. We clarified this in the text.

---

> > ### Author Rebuttal · Reviewer_Ra1S · 2026-04-03
> >
> > - Def 2.2 : does the emptyset works?
> > - I didn't open the url : it is not allowed to put one,so I didn't check the implementation for Fig 2, btu for the approximation I would have like to see a bound.
> > - L1079: didn't prove the two sense of the equivalence : can you provide details ?
> > - for the last point, does the clarification in the text means you remove the "linear-time check" from the paper?
> >
> > Other points are ok for me.

---

> > > ### Author Response · Authors · 2026-04-03
> > >
> > > We thank the reviewer for their continued engagement with our work! We address each follow-up point below.
> > >
> > > > *Def 2.2: does the empty set work?*
> > >
> > > Yes. For $S = \emptyset$: the basis $\underline{S} = \emptyset$ (vacuously, since there are no elements $j \in \emptyset$ to consider), and the closure $\bar{S} = \emptyset$ (since every node $j \in [d]$ is an ancestor of $Y$ by assumption, each $j$ has at least one directed path to $Y$, and no such path can intersect $\emptyset$). The equivalence class is $\{T : \emptyset \subseteq T \subseteq \emptyset\} = \{\emptyset\}$, a singleton class. Its value $\nu(\emptyset) = \mathbb{E}[Y]$ is simply the observational expectation (no intervention), which is always well-defined and identifiable. The empty set is thus both irreducible and closed, and the construction is consistent.
> > >
> > > > *I didn't open the url: it is not allowed to put one...*
> > >
> > > We kindly note that, per the ICML 2026 author response guidelines, anonymous links are explicitly permitted for figures (including tables) and their captions. Quoting the policy: *"links are allowed [...] and links may only be used for figures (including tables) and captions."* **Our link (https://shorturl.at/qFxQf) is anonymous and leads only to supplementary figures/tables.** We respectfully encourage the reviewer to view it, as Rebuttal Figures 1-4 directly address several of the raised concerns.
> > >
> > > Regarding the approximation bound: when $m \geq r$, our estimator covers all equivalence classes and computes the **exact** do-Shapley value (up to machine precision), as shown in Rebuttal Figure 2. When $m < r$, our doEstimator inherits the statistical guarantees of the underlying BaseEstimator. Our structural decomposition reduces the effective number of distinct game values from $2^d$ to $r$, so standard concentration bounds apply with a tighter effective support. We will add a remark to the manuscript making these inherited guarantees explicit.
> > >
> > > > *L1079: didn't prove the two senses of the equivalence: can you provide details?*
> > >
> > > Theorem 5.1 states: $\phi_i$ is identifiable $\iff$ $\forall j \in [d]$, $\nu(\{j\})$ is identifiable. We clarify both directions:
> > >
> > > **All singletons identifiable $\implies$ $\phi_i$ identifiable.** This follows from the contrapositive of the main result proved in Appendix E. We show that if **any** coalition $P_X(Y)$ is non-identifiable, then the hedge $(F, F')$ constructed for $P_X(Y)$ is also a hedge for some singleton $P_{\{s\}}(Y)$, making it non-identifiable (see E.1-E.5 in the appendix and the proof of Theorem 5.1). Contrapositively: if all singletons are identifiable, then all coalition terms $P_S(Y)$ are identifiable, so all $\nu(S)$ are identifiable and so is $\phi_i$.
> > >
> > > **$\phi_i$ identifiable $\implies$ all singletons identifiable.** Let us prove by contrapositive.
> > >
> > > For any singleton $j$, if $P_j(Y)$ were not identifiable, $\nu(j)=E[Y | do(j)]$ would not either, except for a clever canceling of terms in the expectation that rendered it identifiable. However, this invokes a parametric assumption with measure 0 over all possible distributions compatible with the graph. That is why we insist on terming it non-parametric identifiability, to exclude any further assumptions about the distribution. This argument is similar to Pearl’s [1] (page 18, right after Theorem 1.2.4) when discussing the “equivalence” of graph d-separation and conditional independence.
> > >
> > > Hence, if any $P_j(Y)$ is not identifiable, neither is $\nu(j)$. By Eq. 4, $\phi_i$ is a weighted sum over all $r$ equivalence classes. For each singleton $j \in [d]$, let $c_j$ be the class containing the singleton $j$, whose value equals $\nu(j)$ (since $\nu$ is constant within a class). Note that the associated weight $w_i(c_j)$ in Eq. 4 is non-zero since $j$ is a basis of $c_j$. Therefore, $\nu(c_j)$ non-identifiable intervenes in the formula for $\phi_i$ with non-zero weight. By the same argument as before, $\phi_i$ is not identifiable.
> > >
> > > We have updated the paper to clarify this aspect.
> > >
> > > [1] Pearl, J. Causality: Models, Reasoning and Inference. Cambridge University Press, second edition, 2009.
> > >
> > > > *For the last point, does the clarification in the text mean you remove the "linear-time check" from the paper?*
> > >
> > > No; the linear-time identifiability check (Theorem 5.1) remains a key contribution. The clarification in our previous response concerned only the *experiments*: the SCMs learned via GRaSP are Markovian (no unobserved confounders), which trivially guarantees identifiability. We clarified this experimental detail, but the theoretical result is unchanged.
> > >
> > > To clarify terminology: "linear-time check" refers to the fact that the number of ID calls is reduced from $2^d$ (or $r$) to just $d$, one per singleton, which is **linear in the number of variables**. Each individual ID call runs in time polynomial in the graph size. The overall identifiability check thus requires $d \times \text{poly}(|G|)$ time, which is efficient and practical.

---

### Official Review · Reviewer_uUq1 · 2026-03-11

**Soundness:** 3
**Presentation:** 3
**Significance:** 3
**Originality:** 3
**Overall Recommendation:** 4
**Confidence:** 3

**Summary:**

This paper studies efficient algorithms for computing do-Shapley values, which are important for interpreting structural causal models.
The bottleneck is that the Shapley values involve an exponential number of interventions on subsets of causal variables.
A recent work by Parafita et al. (2025) observes that one can reduce the number of interventions by caching values and recognizing that coalitions fall into equivalence classes indexed by irreducible sets (bases). The authors further explore this direction.
The authors develop an efficient algorithm for computing exact Shapley values by identifying all irreducible sets via exploring causal graph structure. The runtime is linear in the number of irreducible sets, which can still be exponential, though it might be small in practice. Thus, the authors propose another algorithm for approximately computing Shapley values given a limited computational budget. Finally, the authors further reduce the identification burden.

**Compliance With Llm Reviewing Policy:**

Affirmed.

**Final Justification:**

The author's rebuttal solved most of my concerns. I therefore keep my positive score.

**Key Questions For Authors:**

1. It seems the parameter r might depend on graph structures. Would it be possible to bound r via some graph parameters (e.g., the longest distance from a node to Y in the graph)?

2. What might be a typical value of budget m in practice?

**Limitations:**

yes

**Strengths And Weaknesses:**

**Strengths:**

1. Do-Shapley is an important way to estimate causal influence in structural causal models, and efficient computation is important because the naive Shapley evaluation requires exponentially many coalition interventions.

2. The paper introduces a clean structural characterization (via irreducible sets and equivalence classes). It reduces the exponential interventions to a computation that scales with the number of irreducible sets.

3. The exact runtime depends on r, the number of irreducible sets (equivalence classes). The paper provides empirical evidence that r can be much smaller on practical graphs (compared to its worst case in theory), making exact computation feasible in many settings.

4. The paper is well written and well organized; the algorithms and the insights are easy to follow.

**Weaknesses:**

1. The runtime can still be exponential in the worst case. In general, exact Shapley computation might be #P-hard.

2. The paper motivates and analyzes the runtime and query complexity of the approximate estimator under limited budget. However, it is less clear what theoretical guarantees can be made about estimation error (e.g., bias, variance).

3. The experiments compare against two model-agnostic estimators (RegressionMSR and LeverageSHAP). However, it would be useful to also compare against the random permutation estimator used in Jung et al. (2022), and the FRA reduction from Parafita et al. (2025).

4. Do-Shapley depends on a specified causal graph. If the graph is wrong, the resulting causal estimations can be misleading.

---

> ### Author Rebuttal · Authors · 2026-03-30
>
> We thank the reviewer for their appreciation of the structure of the paper and our insights into the do-Shapley value, as well as their constructive feedback on limitations and the empirical part of our work.
>
> **Figures and tables for the rebuttal are anonymously available here: https://shorturl.at/qFxQf**
>
> > *runtime can still be exponential in the worst case*
>
> Great point! While exact general Shapley computation notoriously requires $2^d$ evaluations, we show that do-Shapley values can be computed exactly with $r$ evaluations, where $r$ is the number of irreducible sets in a graph. In practice, $r$ is often much smaller than $2^d$. Furthermore, we complement this exact method with **doEstimators that achieve orders of magnitude lower error than SOTA Shapley value estimators employing FRA caching** (Parafita et al., 2025), as shown in Rebuttal Figure 1.
>
> > *The paper motivates and analyzes the runtime and query complexity of the approximate estimator under limited budget. However, it is less clear what theoretical guarantees can be made about estimation error (e.g., bias, variance).*
>
> This is an important question. We added the following discussion to the updated version of the paper:
> If we have sampled all classes, then clearly the doEstimators return the Shapley values to machine precision. If we have *not* sampled all classes, then the theoretical guarantees of the doEstimators depend on the underlying BaseEstimator (e.g., PermutationSHAP, LeverageSHAP, RegressionMSR). Suppose the base estimator can, with $m=O(n/\epsilon^2)$ samples and probability $9/10$, return estimates $\hat{\phi}$, such that
> $$
> \| \phi - \hat{\phi} \|_2^2 \leq \epsilon.
> $$
> BoundarySampler draws classes without replacement proportional to their Shapley weight, then simulates $k \cdot m$ additional samples from discovered classes. This approach avoids further SCM queries (the usual computational bottleneck) and efficiently feeds more information into the BaseEstimator. If these samples were drawn without replacement directly from $\nu$, and $k\cdot m$ did not exceed the sets in the discovered classes, our estimates $\tilde{\phi}$ would satisfy the following with probability $9/10$:
> $$
> \| \phi - \hat{\phi} \|_2^2 \leq \epsilon/\sqrt{k}.
> $$
> We note that because the simulated samples are drawn from discovered (rather than arbitrary) classes, the formal guarantee above serves as an *upper bound on the behavior* rather than a strict certificate. Tightening this analysis to account for the BoundarySampler's importance-weighted discovery order is a promising direction for future work.
>
> > *compare against the random permutation estimator used in Jung et al. (2022), and the FRA reduction from Parafita et al. (2025).*
>
> Thank you for these suggestions. The FRA caching strategy of Parafita et al. (2025) is already employed by all baseline estimators in our experiments; we have clarified this in the updated manuscript! Per your suggestion, we have now also added the PermutationSHAP estimator (used by Jung et al., 2022), likewise with FRA caching. **The results are consistent: doEstimator variants achieve substantially more accurate estimates across all baselines, even when strictly accounting for runtime** (see Rebuttal Figure 1).
>
> > *Do-Shapley depends on a specified causal graph. If the graph is wrong, the resulting causal estimations can be misleading.*
>
> We agree this is a key practical concern. **Our methods correctly compute do-Shapley values for any given SCM; if the graph is misspecified, the values are exact for that graph but may not reflect the true causal structure.** This is analogous to how model-agnostic SHAP faithfully explains a misspecified model. We have added a limitations section discussing this, and note that sensitivity analysis under edge perturbation is a natural direction for future work.
>
> > *r might depend on graph structures…possible to bound r via some graph parameters (e.g., the longest distance from a node to Y in the graph)?*
>
> We find this direction quite appealing! While we do not have tight bounds yet, one easy-to-prove bound is based on the number of parents of the target variable $Y$. Let $p = |\text{Pa}_Y|$. Then:
> $$
> 2^p + (d - p) \leq r \leq 2^d - 2^{d-p} + 1
> $$
> **This shows that $r$ is controlled by $p$, the in-degree of $Y$**: when $Y$ has few parents, $r$ is close to linear in $d$. This holds because $\text{Pa}_Y$ is always a basis with the grand coalition $[d]$ as its closure.
>
> > *What might be a typical value of budget m in practice?*
>
> Good question. The budget $m$ is set by practitioners based on available compute and desired accuracy. Naturally, approaching $m \approx r$ yields highly accurate doEstimator results. Because $r$ is initially unknown for a general graph, practitioners can compute it using AllClasses (optimized for efficiency top-down) or BoundarySampler (less efficient, but discovers classes roughly in order of their "importance" to the do-Shapley values and can be stopped at anytime).

---

> > ### Author Rebuttal · Reviewer_uUq1 · 2026-04-03
> >
> > I thank the authors for their detailed responses. I will keep my positive score.

---

### Official Review · Reviewer_4z3b · 2026-03-13

**Soundness:** 2
**Presentation:** 3
**Significance:** 3
**Originality:** 4
**Overall Recommendation:** 5
**Confidence:** 3

**Summary:**

This paper addresses the computational and identifiability challenges of computing do-Shapley (causal Shapley) values for Structural Causal Models (SCMs). The authors propose partitioning the $2^d$ coalition lattice into $r$ equivalence classes based on each coalition's basis and closure (an irreducible-set view). This allows the Shapley sum to be rewritten as a linear combination over these $r$ classes using analytically computable weights. To leverage this, the paper introduces an exact algorithm that enumerates closed sets and computes do-Shapley values in $O(r(d+e+T))$ time, alongside a structure-aware boundary-sampling estimator that seamlessly transitions to exact computation when the sampling budget $m \geq r$. Finally, the authors make a strong identifiability claim, stating that do-Shapley values are identifiable if and only if the $d$ singleton interventions are identifiable.

**Compliance With Llm Reviewing Policy:**

Affirmed.

**Key Questions For Authors:**

Can you provide a comparison against FRA or a strong caching-only baseline in terms of unique classes found per query, wall-clock runtime, and memory? This would greatly clarify the incremental benefit of the boundary sampler.

**Limitations:**

The authors have not adequately discussed the limitations of their work, particularly regarding the robustness of their methods to graph misspecification. I highly recommend adding a dedicated discussion or sensitivity study showing how randomly perturbed edges impact $r$, accuracy, and runtime.

**Strengths And Weaknesses:**

**Soundness**
I really like the core idea of grouping marginal contributions into basis and closure equivalence classes. It makes intuitive sense and the experiments back up the premise that this drastically reduces the number of terms we need to compute for sparse graphs. the experiments rely entirely on learned models rather than ground truth ones which weakens the causal validity of the whole setup.

**Presentation**
On the presentation front the paper starts strong. The causal blocking examples and the lattice diagrams do a great job of building intuition early on. But the actual reading experience gets frustrating later. The notation also gets confusing because the same letter is overloaded to mean a set, its basis, and its closure all at once.

**Significance**
Scaling causal Shapley values is a well known bottleneck so this paper is tackling a highly relevant problem for the community. If the exact computation method is correct this could be a genuinely useful tool for feature attribution in real world sparse models. I also think the boundary sampling approach is a great practical upgrade over existing estimators. without wall clock runtime comparisons against recent caching methods it is hard to tell just how much of a practical speedup we are actually getting here.

**Originality**
This is definitely the strongest part of the submission. The idea to compress the Shapley sum down to a smaller set of equivalence classes with analytic weights is a very creative and elegant angle on a familiar problem. It feels like a deep structural insight rather than just a minor algorithmic tweak. I also really appreciated how the authors connected their framework to other probabilistic values. It shows they have a thorough understanding of the problem space and combining this decomposition with a targeted sampling strategy is a great novel contribution.

---

> ### Author Rebuttal · Authors · 2026-03-30
>
> We thank the reviewer for their appreciation of our insights into the structure of the do-Shapley value, and constructive feedback on improving the empirical part of our work.
>
> **Figures and tables for the rebuttal are anonymously available here: https://shorturl.at/qFxQf**
>
> > *the experiments rely entirely on learned models rather than ground truth ones which weakens the causal validity of the whole setup.*
>
> Thank you for highlighting this point! Because plentiful, high-quality ground-truth SCMs are scarce, our primary experimental pipeline learns SCMs from the TALENT datasets using SOTA methods like GRaSP. Nonetheless, to address your concern, **we have run additional experiments on ground-truth SCMs from the `pgmpy` library** (asia, sachs, survey, alarm, child, insurance, water; see Rebuttal Figures 3 and 4). While these are fewer in number, **the results are consistent with our findings on learned SCMs**, supporting the generality of our approach.
>
> > *On the presentation front the paper starts strong. The causal blocking examples and the lattice diagrams do a great job of building intuition early on. But the actual reading experience gets frustrating later. The notation also gets confusing because the same letter is overloaded to mean a set, its basis, and its closure all at once.*
>
> We appreciate the kind words about the introduction. We agree that notation density increases in later sections. Our choice to use $S$, $\underline{S}$, and $\bar{S}$ was deliberate; we believed variations of the same letter would impose less cognitive load than unrelated letters (e.g., $S$, $B$, $C$). That said, **we will add a notation summary table and clarify variable reuse in later Sections (e.g. 3 and 5)** to improve readability. We welcome any further specific suggestions!
>
> > *without wall clock runtime comparisons against recent caching methods it is hard to tell just how much of a practical speedup we are actually getting here.*
>
> Great point! The plots in the original version of the paper showed MSE by number of queries, but it’s possible (and actually the case) that our doEstimators take longer to run per query because they explore the graph. We have added a comparison of MSE by time for each estimator, shown in Rebuttal Figures 1 and 3. This plot fairly shows error by total time, and the findings suggest that the **doEstimators, in the same amount of time, are more accurate than their standard variants by up to 10 orders of magnitude**.
>
> > *Can you provide a comparison against FRA or a strong caching-only baseline in terms of unique classes found per query, wall-clock runtime, and memory? This would greatly clarify the incremental benefit of the boundary sampler.*
>
> Great question. The FRA caching strategy of Parafita et al. (2025) is *already employed* by the standard estimators in our experiments. We have made this clear in the updated manuscript. In addition, we highlight the number of classes found per query and per second in Rebuttal Tables 1 through 4.
>
> While the standard Shapley value estimators find more classes per second and their doEstimator variants find more classes per query, the efficiency of both types of algorithm are comparable. Really, the meaningful difference between them, as shown in Rebuttal Figures 1 and 3, is that the doEstimator variants can exploit the irreducible structure of causal graphs to exactly recover the Shapley values. To put it another way, **both kinds of algorithms are roughly getting the same amount of information per second/query, but the doEstimators can leverage that information for much more accurate estimates**.
>
> > *The authors have not adequately discussed the limitations of their work, particularly regarding the robustness of their methods to graph misspecification. I highly recommend adding a dedicated discussion or sensitivity study showing how randomly perturbed edges impact $r$, accuracy, and runtime.*
>
> Thank you for bringing up this important limitation! We agree that graph misspecification is a key practical concern. **Our methods correctly compute do-Shapley values for *any* given SCM; if the graph is misspecified, the returned values are exact for that (incorrect) graph.** This is analogous to how standard SHAP values are exact for a given model even if the model itself is misspecified. We have made this clear in the limitations section, and believe that your suggestion of investigating sensitivity to edge perturbations is a promising direction for future work.

---

> > ### Author Rebuttal · Reviewer_4z3b · 2026-04-03
> >
> > i will keep my positive score

---

### Decision · Program_Chairs · 2026-04-30

**Decision:**

Accept (regular)

**Comment:**

The paper addresses the issue of combinatorial explosion of terms in Shapley value based causal contribution in an elegant way.
The contribution is transparent, theoretically sound, and the authors have adequately responded to questions on practical relevance given that exact computation of Shapley values may not be required in practice (and thus not be a fair baseline).